# When Do Flat Minima Optimizers Work?

**Jean Kaddour**[*]
Centre for Artificial Intelligence
University College London

**Linqing Liu**[*]
Centre for Artificial Intelligence
University College London

**Ricardo Silva**
Department of Statistical Science
University College London

**Matt J. Kusner**
Centre for Artificial Intelligence
University College London

## Abstract

Recently, *flat-minima optimizers*, which seek to find parameters in low-loss neighborhoods, have been shown to improve a neural network's generalization performance over stochastic and adaptive gradient-based optimizers. Two methods have received significant attention due to their scalability: 1. Stochastic Weight Averaging (SWA), and 2. Sharpness-Aware Minimization (SAM). However, there has been limited investigation into their properties and no systematic benchmarking of them across different domains. We fill this gap here by comparing the loss surfaces of the models trained with each method and through broad benchmarking across computer vision, natural language processing, and graph representation learning tasks. We discover several surprising findings from these results, which we hope will help researchers further improve deep learning optimizers, and practitioners identify the right optimizer for their problem.

## 1  Introduction

Stochastic gradient descent (SGD) methods are central to neural network optimization [6]. Recently, one class of algorithms has focused on biasing SGD methods towards so-called '*flat*' minima, which are located in large weight space regions with very similar low loss values [43]. Theoretical and empirical studies [21, 77, 9, 55, 49, 5, 12] postulate that such flatter regions generalize better than sharper minima, e.g., due to the flat minimizer's robustness against loss function shifts between train and test data, as illustrated in Fig. 1. Two popular flat-minima optimization approaches are: 1. Stochastic Weight Averaging (SWA) [48], and 2. Sharpness-Aware Minimization (SAM) [22].

While both strategies aim to find flatter minima, they operate much differently. On the one hand, SWA is based on the intuition that, near a flat minimum, gradients are smaller, leaving many iterates in that flat region. Therefore, averaging iterates will produce a solution that is pulled towards these flatter regions, see Fig. 1, top. On the other hand, SAM minimizes the maximum loss around a neighborhood of the current iterate. This way, a region around the iterate is designed to have uniformly low loss; see Fig. 1, bottom. Crucially, SAM requires an additional forward/backward pass for each parameter update, making it more expensive than SWA.

Despite the successes [76, 3, 51, 12, 4] of SWA and SAM in some domains, we are unaware of a systematic comparison between them that would help practitioners to choose the right optimizer for their problem and researchers to develop better optimizers. The SWA [48] paper was published in 2018, and the SAM [22] paper in 2021; however, the SAM paper, and its most noticeable follow-ups [65, 12, 103], do not compare against SWA. Further, there is very limited overlap in

---

[*]Equal contribution, correspondence to {jean.kaddour,linqing.liu}.20@ucl.ac.uk

36th Conference on Neural Information Processing Systems (NeurIPS 2022).

terms of the model architecture and dataset used in the experiments among both papers, which are likely further confounded by other differences in the training procedures (e.g. data augmentations, hyper-parameters, etc.).

**Contributions**

1. **In-depth comparison of minima found by SWA and SAM:** We visualize linear interpolations between different models and quantify the minimizers' flatnesses. This analysis yields 4 insights, e.g., despite SAM finding flatter solutions than SWA as quantified by Hessian eigenvalues, they can be close to sharp directions, a phenomenon that has been overlooked in the previous SAM literature. Averaging SAM iterates leads to the flattest among all minima.

2. **Rigorous comparison of SWA and SAM's performance over 42 tasks:** We empirically compare the optimizers with a rigorous model selection procedure on a broad range of tasks across different domains (CV, NLP, and GRL), model types (MLPs, CNNs, Transformers) and tasks (classification, self-supervised learning, open-domain question answering, natural language understanding, and node/graph/link property prediction). We discuss 9 findings, e.g., that both dataset and architecture impact their effectiveness, that for NLP tasks, SAM improves over SWA in most cases, and that the converse holds for GRL tasks. When flat-minima optimizers do not help, we notice clear discrepancies between the shapes of loss and accuracy curves. To assist future work, we open-source the code for all pipelines and hyper-parameters to reproduce the results.

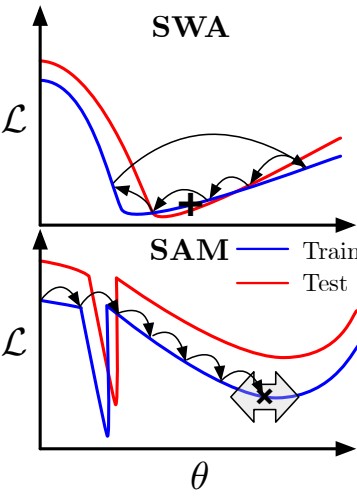

Figure 1: The mechanics behind SWA and SAM, whose solution is denoted by $+$ and $\times$, respectively. SWA produces a solution $\theta$ that is pulled towards flatter regions, while SAM approximates sharpness within the parameters' neighborhood (arrows).

## 2 Background and Related Work

### 2.1 Stochastic Gradient Descent (SGD)

The classic optimization framework of machine learning is empirical risk minimization

$$\mathcal{L}\left(\boldsymbol{\theta}\right) = \frac{1}{N} \sum_{i=1}^{N} \ell\left(\boldsymbol{x}_i; \boldsymbol{\theta}\right) \tag{1}$$

where $\boldsymbol{\theta} \in \mathbb{R}^d$ is a vector of parameters, $\{\boldsymbol{x}_1, \ldots, \boldsymbol{x}_N\}$ is a training set of inputs $\boldsymbol{x}_n \in \mathbb{R}^D$, and $\ell(\boldsymbol{x}; \boldsymbol{\theta})$ is a loss function quantifying the performance of parameters $\boldsymbol{\theta}$ on $\boldsymbol{x}$. SGD samples a minibatch $\mathcal{S} \subset \{1, \ldots, N\}$ of size $|\mathcal{S}| \ll N$ from the training set and updates the parameters through

$$\boldsymbol{\theta}_{t+1}^{\mathrm{SGD}} = \boldsymbol{\theta}_t - \eta \boldsymbol{g}\left(\boldsymbol{\theta}_t\right), \text{ where } \boldsymbol{g}\left(\boldsymbol{\theta}\right) = \frac{1}{|\mathcal{B}|} \sum_{i \in \mathcal{B}} \nabla \ell\left(\boldsymbol{\theta}; \boldsymbol{x}_i\right), \tag{2}$$

for a length specified by $\eta$, the learning rate.

### 2.2 Stochastic Weight Averaging (SWA)

The idea of averaging weights dates back to accelerating the convergence speed of SGD [78, 51]. SWA's motivation is based on the following observation about SGD's behavior when training neural networks: it often traverses regions of the weight space that correspond to high-performing models but rarely reaches the central points of this optimal set. Averaging the parameter values over iterations moves the solution closer to the centroid of this space of points.

The SWA update rule is the cumulative moving average

$$\boldsymbol{\theta}_{t+1}^{\mathrm{SWA}} \leftarrow \frac{\boldsymbol{\theta}_t^{\mathrm{SWA}} \cdot l + \boldsymbol{\theta}_t^{\mathrm{SGD}}}{l + 1}, \tag{3}$$

| **Algorithm 1** Stochastic Weight Averaging [48] | **Algorithm 2** Sharpness-Aware Minimization [22] |
|---|---|
| **Input:** Loss function $\mathcal{L}$, training budget in number of iterations $b$, training dataset $\mathcal{D} := \cup_{i=1}^{n}\{\boldsymbol{x}_i\}$, mini-batch size $|\mathcal{B}|$, averaging start epoch $E$, averaging frequency $\nu$, (scheduled) learning rate $\eta$, initial weights $\boldsymbol{\theta}_0$. | **Input:** Loss function $\mathcal{L}$, training budget in number of iterations $b$, training dataset $\mathcal{D} := \cup_{i=1}^{n}\{\boldsymbol{x}_i\}$, mini-batch size $|\mathcal{B}|$, neighborhood radius $\rho$, (scheduled) learning rate $\eta$, initial weights $\boldsymbol{\theta}_0$. |
| **for** $k \leftarrow 1, \dots, b$ **do** | **for** $k \leftarrow 1, \dots, b$ **do** |
|   Sample a mini-batch $\mathcal{B}$ from $\mathcal{D}$ |   Sample a mini-batch $\mathcal{B}$ from $\mathcal{D}$ |
|   Compute gradient $\boldsymbol{g} \leftarrow \nabla\mathcal{L}(\boldsymbol{\theta}_t)$ |   Compute worst-case perturbation $\widehat{\boldsymbol{\epsilon}} \leftarrow \rho \dfrac{\nabla\mathcal{L}(\boldsymbol{\theta})}{\|\nabla\mathcal{L}(\boldsymbol{\theta})\|_2}$ |
|   Update parameters $\boldsymbol{\theta}_{t+1} \leftarrow \boldsymbol{\theta}_t - \eta\boldsymbol{g}$ |   Compute gradient $\boldsymbol{g} \leftarrow \nabla\mathcal{L}\left(\boldsymbol{\theta}_t^{\text{SAM}}+\widehat{\boldsymbol{\epsilon}}\right)$ |
|   **if** $k \geq E$ and $\mathrm{mod}(k,\nu) = 0$ **then** |   Update parameters $\boldsymbol{\theta}_{t+1}^{\text{SAM}} \leftarrow \boldsymbol{\theta}_t^{\text{SAM}} - \eta\boldsymbol{g}$ |
|     $\boldsymbol{\theta}_{t+1}^{\text{SWA}} = \left(\boldsymbol{\theta}_t^{\text{SWA}} \cdot l + \boldsymbol{\theta}_{t+1}^{\text{SWA}}\right) / (l+1)$ | **end for** |
|   **end if** | **return** $\theta^{\text{SAM}}$ |
| **end for** | |
| **return** $\theta^{\text{SWA}}$ | |

where $l$ is the number of distinct parameters averaged so far and $t$ is the SGD iteration number.[2]

SWA has two hyper-parameters: the update frequency $\nu$ and starting epoch $E$. When using a constant learning rate, Izmailov et al. [48] suggests updating the parameters once after each epoch, i.e., $\nu \approx \frac{N}{|\mathcal{B}|}$, and starting at $E \approx 0.75T$, where $T$ is the training budget required to train the model until convergence with conventional SGD training.

He et al. [39] argue that SWA may always improve generalization, regardless of the loss function's geometry. Kaddour [51] show that averaging a specific range of weights can speed up training convergence. Cha et al. [8] argue that tuning $\nu$ and $E$ carefully is necessary to make it work effectively in domain generalization (DG) tasks. Besides DG tasks, a list of tuned hyper-parameters based on a fair model selection procedure across different architectures and tasks has been missing in the literature. To the best of our knowledge, Cha et al. [8] is the only study that compares SWA and SAM over the same experiments, but it focuses on domain generalization tasks which we, therefore, leave out in this work.

## 2.3 Sharpness-Aware Minimization (SAM)

While SWA is implicitly biased towards flat minima, SAM *explicitly* approximates the flatness around parameters $\boldsymbol{\theta}$ to guide the parameter update. It first computes the worst-case perturbation $\boldsymbol{\epsilon}$ that maximizes the loss within a given neighborhood $\rho$, then minimizes the loss w.r.t. the perturbed weights $\boldsymbol{\theta} + \boldsymbol{\epsilon}$. Formally, SAM finds $\boldsymbol{\theta}$ by solving the minimax problem:

$$\min_{\boldsymbol{\theta}} \max_{\|\boldsymbol{\epsilon}\|_2 \leq \rho} \mathcal{L}(\boldsymbol{\theta} + \boldsymbol{\epsilon}), \tag{4}$$

where $\rho \geq 0$ is a hyperparameter.

To find the worst-case perturbation $\boldsymbol{\epsilon}^*$ efficiently in practice, Foret et al. [22] approximates Eq. (4) via a first-order Taylor expansion of $\mathcal{L}(\boldsymbol{\theta} + \boldsymbol{\epsilon})$ w.r.t. $\boldsymbol{\epsilon}$ around $\mathbf{0}$, obtaining

$$\boldsymbol{\epsilon}^* \approx \underset{\|\boldsymbol{\epsilon}\|_2 \leq \rho}{\arg\max} \, \boldsymbol{\epsilon}^\top \nabla_{\boldsymbol{\theta}}\mathcal{L}(\boldsymbol{\theta}) \approx \underbrace{\rho \cdot \frac{\nabla_{\boldsymbol{\theta}}\mathcal{L}(\boldsymbol{\theta})}{\|\nabla_{\boldsymbol{\theta}}\mathcal{L}(\boldsymbol{\theta})\|}}_{=:\widehat{\boldsymbol{\epsilon}}}. \tag{5}$$

In words, $\widehat{\boldsymbol{\epsilon}}$ is simply the scaled gradient of the loss function w.r.t to the current parameters $\boldsymbol{\theta}$. Given $\widehat{\boldsymbol{\epsilon}}$, the altered gradient used to update the current $\boldsymbol{\theta}_t$ (in place of $\boldsymbol{g}(\boldsymbol{\theta}_t)$) is

$$\nabla_{\boldsymbol{\theta}} \max_{\|\boldsymbol{\epsilon}\|_2 \leq \rho} \mathcal{L}(\boldsymbol{\theta} + \boldsymbol{\epsilon}) \approx \nabla_{\boldsymbol{\theta}}\mathcal{L}(\boldsymbol{\theta})|_{\boldsymbol{\theta}+\widehat{\boldsymbol{\epsilon}}}.$$

Due to Eq. (5), SAM's computational overhead consists of an additional forward and backward pass per parameter update step compared to SWA and non-flat optimizers.

---

[2] SWA parameters are constant between averaging steps.

SAM's performance strongly depends on the neighborhood radius $\rho$. For example, Chen et al. [12], Wu et al. [93] show that $\rho$ should be set to values outside the originally considered ranges by Foret et al. [22]. Analogously to Sec. 2.2, this lack of coherence among hyper-parameter tuning protocols in the SAM literature makes it tricky to determine SAM's comparative effectiveness.

## 2.4 Other Flat-Minima Optimizers

There are several extensions of SWA [36, 8] and SAM [65, 103, 101]. For simplicity, we do not consider them in this work. Besides SWA and SAM, other flat-minima optimizers include e.g., [9, 84]. However, due to their computational cost and/or lack of performance gains, we do not include them in this work. Chaudhari et al. [9] requires $[5, 20]$ forward and backward passes per parameter update. Sankar et al. [84] similarly requires $[5, 10]$ forward and backward passes to estimate the Hessian trace and 6 of 7 experiments yield minimal improvement of $\leq 0.27\%$, see Table 1 in Sankar et al. [84]. In contrast, SWA and SAM have been shown to increase performance by multiple percentage points in some cases [8, 12] while requiring fewer computational resources.

## 3  How do minima found by SWA and SAM differ?

In this section, we investigate SWA and SAM solutions in two prototypical deep learning tasks, where these optimizers improve over the baseline. Our goal is to understand better their geometric properties (instead of their generalization performance, which is the focus of Sec. 4).

First, we investigate the behavior of the loss landscape along the line between non-flat and flat solutions (Sec. 3.1). Previous studies successfully used such linear interpolations to gain novel insights, e.g., for training dynamics [32, 25], regularization [69, 28], and network pruning [26]. Second, motivated by findings in Sec. 3.1, we average SAM iterates and visualize interpolations between averaged and non-averaged solutions (Sec. 3.2). Interestingly, the averaged SAM solution is less susceptible to asymmetric directions. Third, we compare quantitative measurements of all solutions' flatnesses (Sec. 3.3). Here, we compute dominant Hessian eigenvalues, as commonly used in the flat minima literature [9, 98, 12, 22]. Lastly, in Appendix A.1, we further compute CKA [61] and cosine similarities between SWA/SAM's network output logits.

We choose the following two disparate learning settings: (i) a well-known image classification task, widely used for evaluation in flat-minima optimizer papers, and (ii) a novel, challenging Python code summarization task, on which state-of-the-art models achieve only around $16\%$ F1 score on the test set (which is **higher** than its commonly achieved accuracy on the more challenging training set), and that has not been explored yet in the flat-minima literature. Specifically, for (i), we investigate the loss/accuracy surfaces of a WideResNet28-10 [99] model on CIFAR-100 [63] (baseline non-flat optimizer: SGD with momentum (SGD-M)) [83]. For (ii), we use the theoretically-grounded Graph Isomorphism Network [95] model on OGB-Code2 [45] (baseline optimizer: Adam [56]).

All optimizers start from the same initialization. We denote the minimizer produced by the non-flat methods (SGD-M and Adam) by $\theta^{\text{NF}}$ and the flat ones by $\theta^{\text{SWA}}$ and $\theta^{\text{SAM}}$.

### 3.1  What is between non-flat and flat solutions?

We start by comparing the similarity of flat and non-flat minimizers through linear interpolations. This analysis allows us to understand if they are in the same basin and how close they are to a region of sharply-increasing loss, where we expect loss/accuracy to differ widely between train and test. Further, for each of our four observations, we recommend a future work direction.

To linearly interpolate between two sets of parameters $\theta$ and $\theta'$, we parameterize the line connecting these two by choosing a scalar parameter $\alpha$ and defining the weighted average $\theta(\alpha) = (1-\alpha)\theta + \alpha\theta'$. If there exists no high-loss barrier between two networks $\theta, \theta'$ along the linear interpolation, we say that they are located in the same *basin*, i.e., $\{\theta, \theta'\} \in \Omega$. [75, 102]. A basin is an area in the parameter space where the loss function has relatively low values. Due to NN non-linearities, the linear combination of the weights of two accurate models does not necessarily define an accurate model. Hence, we generally expect high-loss barriers along the linear interpolation path.

While there are alternative distance measures that could be used to compare two networks, they typically either (a) do not offer clear interpretations, as pointed out by Frankle et al. [26], or (b) yield

trivial network connectivity results, such as *non-linear* low-loss paths, which can be found for any two network minimizers [20, 27, 33, 23].

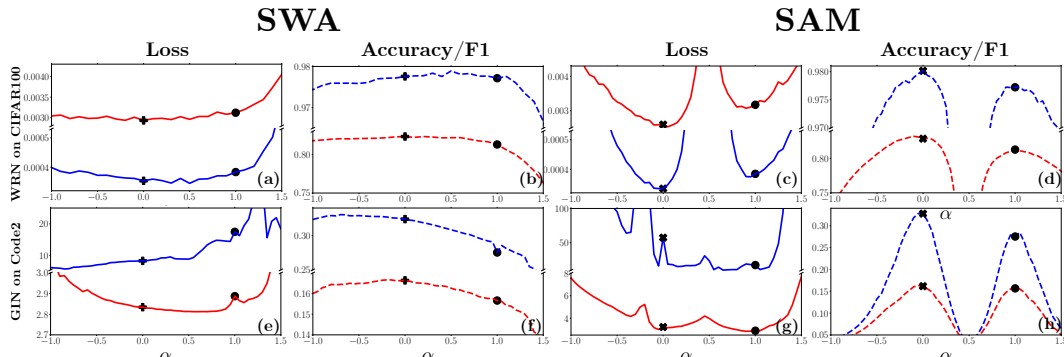

Figure 2: Training (blue) and test (red) losses (—) and accuracies (·····) of linear interpolations $\boldsymbol{\theta}(\alpha) = (1 - \alpha)\boldsymbol{\theta} + \alpha\boldsymbol{\theta}'$ (for $\alpha \in [-1, 1.5]$) between SWA (+) and SAM (×) solutions ($\alpha = 0.0$) and non-flat baseline solutions ($\bullet, \alpha = 1.0$).

**Obs. 1:** $\{\boldsymbol{\theta}^{\mathbf{SWA}}, \boldsymbol{\theta}^{\mathbf{NF}}\} \in \boldsymbol{\Omega}^{\mathbf{NF}}$. $\boldsymbol{\theta}^{\mathrm{SWA}}$ and $\boldsymbol{\theta}^{\mathrm{NF}}$ are in the same basin, as can be seen in Figures 2a and 2e. Additionally, $\boldsymbol{\theta}^{\mathrm{NF}}$ is near the periphery of a sharp increase in loss, as can be seen when moving in the direction from $\boldsymbol{\theta}^{\mathrm{SWA}}$ to $\boldsymbol{\theta}^{\mathrm{NF}}$ (i.e., $\alpha > 1$). Conversely, $\boldsymbol{\theta}^{\mathrm{SWA}}$ finds flat regions that change slowly in the loss. This bias of SWA to flatter loss beneficially transfers to the accuracy landscape too: Figures 2b and 2f show the accuracy/F1 score rapidly dropping off approaching and beyond $\boldsymbol{\theta}^{\mathrm{NF}}$. Interestingly, in Figures 2e and 2f, we see that for Code2, for $\alpha < 0$, there exist solutions with even better training loss/accuracy but worse test loss/accuracy. However, $\boldsymbol{\theta}_{\mathrm{GIN}}^{\mathrm{SWA}}$ is close to the test accuracy maximizer along this interpolation. Future work may inspect why the cross entropy loss function used for GIN/Code2 seems less well correlated with its accuracy compared to WRN/CIFAR100.

**Obs. 2:** $\boldsymbol{\theta}^{\mathbf{SAM}} \in \boldsymbol{\Omega}^{\mathbf{SAM}} \neq \boldsymbol{\Omega}^{\mathbf{NF}}$. $\boldsymbol{\theta}^{\mathrm{SAM}}$ and $\boldsymbol{\theta}^{\mathrm{NF}}$ are not in the same basin: Figures 2c and 2g show that there is a high loss barrier between them, respectively. Figures 2d and 2h show that $\boldsymbol{\theta}^{\mathrm{SAM}}$ and even nearby points in parameter space achieve higher accuracies/F1 scores (i.e., generalize better) than $\boldsymbol{\theta}^{\mathrm{NF}}$ and points around it. This is an interesting result because we expect different basins to produce qualitatively different predictions, one of the motivations behind combining models, even if they exhibit different performances [46, 67]. Grewal & Bui [34] successfully combine models yielded by different optimizers, and we think future work should study ensembling SAM and non-SAM solutions.

**Obs. 3: SAM finds a saddle point.** Figure 2g shows $\boldsymbol{\theta}_{\mathrm{GIN}}^{\mathrm{SAM}}$ being located in a sharp training loss minimum whose loss is much higher than $\boldsymbol{\theta}^{\mathrm{NF}}$. Yet, its test loss is slightly higher, and its F1 score is better. We visualize 2D plots moving along random directions (not shown here due to space) to confirm that $\boldsymbol{\theta}_{\mathrm{GIN}}^{\mathrm{SAM}}$ is a saddle point (Appendix A.2). A common pathology among curvature-based methods is that they attract saddle points [16]. Since SAM takes some form of curvature into account, too, we believe that future work should investigate SAM's propensity to find saddle points and potential remedies.

**Obs. 4:** $\boldsymbol{\theta}^{\mathbf{SAM}}$ **is closer to sharper directions than** $\boldsymbol{\theta}^{\mathbf{SWA}}$, as can be seen by $\mathcal{L}_{\mathrm{tr/te}}(\boldsymbol{\theta}^{\mathrm{SAM}}(0.1)) \approx 2 \cdot \mathcal{L}_{\mathrm{tr/te}}(\boldsymbol{\theta}^{\mathrm{SAM}}(-0.1))$, while $\mathcal{L}_{\mathrm{tr/te}}(\boldsymbol{\theta}^{\mathrm{SWA}}(0.1)) \approx \mathcal{L}_{\mathrm{tr/te}}(\boldsymbol{\theta}^{\mathrm{SWA}}(-0.1))$, where $\mathcal{L}(\cdot)_{\mathrm{tr/te}}$ refers to both training and test loss functions. A possible explanation for SAM being closer to sharp sides is that while it finds different basins than SGD/SWA by smoothing the loss surface (as illustrated in Fig. 1), *within* a local basin, it may oscillate around the minimizer similarly as SGD. One cause for this can be that $\boldsymbol{\Omega}^{\mathrm{SAM}}$'s hypersphere is larger than SAM's radius $\rho$. If that holds, then given a small enough learning rate, we expect it to oscillate around $\boldsymbol{\theta}^* \in \boldsymbol{\Omega}^{\mathrm{SAM}}$ (the smaller the learning rate, the less likely it escapes the basin due to that stochasticity). Two possible remedies are: (1) adapt/schedule $\rho$, or (2) average SAM iterates to bias its solution towards the flatter side. (1) has been explored by [103, 101]. We try (2) in the next subsection. Future work may study SAM's basin escape time, e.g., using convolutions [58] or stochastic differential equations [102].

## 3.2 What happens if we average SAM iterates?

Based on observation 4: "$\theta^{\text{SAM}}$ **is closer to sharper directions than** $\theta^{\text{SWA}}$", averaging SAM iterates may further improve generalization, referred to as *Weight-Averaged Sharpness-Aware Minimization* (WASAM). The reason is that while SAM finds better-performing basins, *within* the basin, its final iterate may still be near a side that increases sharply in the loss.

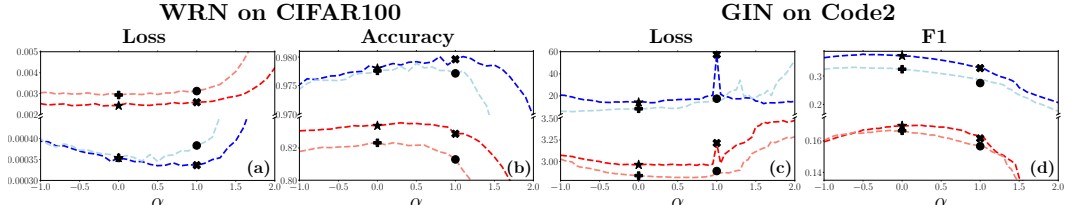

Figure 3: Training (blue) / test (red) losses (—) / accuracies (·····) between non-flat baseline (●) ↔ SWA (+), SAM (×) ↔ WASAM (⋆).

Starting with the first of the two previously analyzed settings (WRN/CIFAR100), Figures 3a, and 3b show that $\theta^{\text{SWA+SAM}}_{\text{WRN}}$ (marker: ⋆) achieves the lowest test loss and highest test accuracy, respectively. What stands out in comparison to the previous plots is $\theta^{\text{SAM}}_{\text{WRN}}$'s (×) proximity to sharp sides, surprisingly similar to $\theta^{\text{NF}}_{\text{WRN}}$ (●) here and in Figures 2c and 2e. As we hoped, $\theta^{\text{SWA+SAM}}_{\text{WRN}}$ is indeed closer to a flatter region, as can be seen by $\mathcal{L}_{\text{tr/te}}(\theta^{\text{SWA+SAM}}_{\text{WRN}}(-0.2)) \approx \mathcal{L}_{\text{tr/te}}(\theta^{\text{SWA+SAM}}_{\text{WRN}}(0.2))$.

In GIN/OGB-Code2, one unanticipated finding is that $\theta^{\text{SWA+SAM}}_{\text{GIN}}$ escapes the (previously discussed) saddle point of $\theta^{\text{SAM}}_{\text{GIN}}$, appearing here as a maximum in Figure 3c. A likely reason is that SAM traversed nearby flatter regions before arriving at the saddle point, especially if it is a non-strict saddle. In terms of F1 score, Figure 3d shows that while $\theta^{\text{SWA}}_{\text{GIN}}$ (+) and $\theta^{\text{SAM}}_{\text{GIN}}$ perform about equally well, the flatter region found by $\theta^{\text{SWA+SAM}}_{\text{GIN}}$ improves over both.

## 3.3 How "flat" are the found minima?

We now quantify the flatnesses of all four optimizers over both tasks by computing the median of the dominant Hessian eigenvalue across all training set batches using the Power Iteration algorithm [74, 98]. This metric measures the worst-case loss landscape curvature. We choose this metric as it is very commonly used in the minima flatness literature, e.g., [9, 12, 22, 97, 18, 62, 85].

Table 1 shows that SAM leads to flatter minima than SWA in both cases. Interestingly $\lambda_{\max}(\theta^{\text{NF}}_{\text{WRN}}) \approx 2.5 \cdot \lambda_{\max}(\{\theta^{\text{SWA}}, \theta^{\text{SAM}}\})$, while $\lambda_{\max}(\theta^{\text{NF}}_{\text{WRN}}) \approx 5.75\lambda_{\max}(\theta^{\text{SWA+SAM}}_{\text{WRN}})$, indicating room for improvement in terms of flatness for both SWA and SAM. The relative differences are less dramatic for GIN/Code2, although surprisingly $\lambda_{\max}(\theta^{\text{NF}}_{\text{GIN}}) \approx \lambda_{\max}(\theta^{\text{SWA}}_{\text{GIN}})$. In sum, averaging SAM iterates leads to the flattest minima **and** best-performing minima in both cases (see Sec. 4).

Table 1: Median $\lambda_{\max}$ of Hessian over all training set batches.

| Task | Baseline | SWA | SAM | WASAM |
|---|---|---|---|---|
| WRN on CIFAR100 | 673 | 265 | 237 | **117** |
| GIN on Code2 | 16.65 | 16.79 | 11.31 | **9.96** |

# 4 How do SWA and SAM perform on a broad set of experiments?

As we point out in the introduction, there is almost no overlap and consistency regarding reported SWA and SAM results in the literature. This section addresses this gap. For example, Bahri et al. [4], Chen et al. [12] illustrate that the flat minima found by SAM improve generalization on Transformer [90] architectures compared to non-flat optimizers, but they do not compare against SWA. Hence, it is unclear if the computationally cheaper SWA may provide better or similar performance.

We compare flat minimizers SWA, SAM, and averaged SWA iterates (WASAM) over the non-flat minimizers across a range of different tasks in the domains of computer vision, natural language processing, and graph representation learning. We average all runs at least three times across random

Table 2: CV test results: Supervised Classification (SC), and Self-Supervised Learning (SSL) tasks.

| Task | Model | Baseline | SWA | SAM | WASAM |
|------|-------|----------|-----|-----|-------|
| SC: CIFAR10 | WRN-28-10 | $96.78_{\pm 0.03}$ | $-0.05_{\pm 0.04}$ | $+\mathbf{0.34}_{\pm 0.09}$ | $+0.25_{\pm 0.05}$ |
| | PN-272 | $96.73_{\pm 0.14}$ | $+0.22_{\pm 0.14}$ | $+\mathbf{0.42}_{\pm 0.06}$ | $+\mathbf{0.41}_{\pm 0.02}$ |
| | ViT-B-16 | $98.95_{\pm 0.02}$ | $-0.04_{\pm 0.04}$ | $+0.07_{\pm 0.01}$ | $+\mathbf{0.10}_{\pm 0.01}$ |
| | Mixer-B-16 | $96.65_{\pm 0.03}$ | $+0.02_{\pm 0.03}$ | $+0.19_{\pm 0.05}$ | $+\mathbf{0.22}_{\pm 0.06}$ |
| SC: CIFAR100 | WRN-28-10 | $80.93_{\pm 0.19}$ | $+1.62_{\pm 0.06}$ | $+1.82_{\pm 0.14}$ | $+\mathbf{2.24}_{\pm 0.14}$ |
| | PN-272 | $80.86_{\pm 0.12}$ | $+1.88_{\pm 0.04}$ | $+2.33_{\pm 0.08}$ | $+\mathbf{2.60}_{\pm 0.09}$ |
| | ViT-B-16 | $92.77_{\pm 0.07}$ | $-0.12_{\pm 0.05}$ | $+\mathbf{0.19}_{\pm 0.09}$ | $+\mathbf{0.13}_{\pm 0.07}$ |
| | Mixer-B-16 | $83.77_{\pm 0.08}$ | $+0.45_{\pm 0.06}$ | $+0.52_{\pm 0.15}$ | $+\mathbf{0.97}_{\pm 0.12}$ |
| SSL: CIFAR10 | MoCo | $\mathbf{89.25}_{\pm \mathbf{0.07}}$ | $-0.03_{\pm 0.10}$ | $-0.25_{\pm 0.06}$ | $-0.17_{\pm 0.10}$ |
| | SimCLR | $\mathbf{88.66}_{\pm \mathbf{0.08}}$ | $-0.05_{\pm 0.06}$ | $+0.05_{\pm 0.04}$ | $-0.13_{\pm 0.06}$ |
| | SimSiam | $\mathbf{89.86}_{\pm \mathbf{0.22}}$ | $+0.12_{\pm 0.26}$ | $+0.07_{\pm 0.10}$ | $+0.11_{\pm 0.10}$ |
| | BarlowTwins | $\mathbf{86.34}_{\pm \mathbf{0.24}}$ | $-0.09_{\pm 0.19}$ | $+0.09_{\pm 0.15}$ | $+0.14_{\pm 0.05}$ |
| | BYOL | $90.32_{\pm 0.14}$ | $+\mathbf{0.70}_{\pm 0.05}$ | $+0.14_{\pm 0.03}$ | $+0.21_{\pm 0.07}$ |
| | SwaV | $87.28_{\pm 0.05}$ | $+\mathbf{0.09}_{\pm 0.06}$ | $+0.07_{\pm 0.12}$ | $+0.02_{\pm 0.06}$ |
| SSL: ImageNette | MoCo | $81.74_{\pm 0.18}$ | $+0.97_{\pm 0.10}$ | $+0.91_{\pm 0.32}$ | $+\mathbf{1.40}_{\pm 0.10}$ |
| | SimCLR | $83.28_{\pm 0.22}$ | $+0.95_{\pm 0.25}$ | $+0.18_{\pm 0.24}$ | $+\mathbf{1.07}_{\pm 0.13}$ |
| | SimSiam | $81.77_{\pm 0.14}$ | $+0.20_{\pm 0.37}$ | $+\mathbf{0.33}_{\pm 0.28}$ | $+0.18_{\pm 0.26}$ |
| | BarlowTwins | $77.49_{\pm 0.36}$ | $+0.20_{\pm 0.16}$ | $+0.47_{\pm 0.27}$ | $+\mathbf{0.66}_{\pm 0.57}$ |
| | BYOL | $84.16_{\pm 0.14}$ | $+\mathbf{0.76}_{\pm 0.08}$ | $+0.15_{\pm 0.25}$ | $+0.31_{\pm 0.19}$ |
| | SwaV | $88.16_{\pm 0.31}$ | $+\mathbf{1.04}_{\pm 0.27}$ | $+0.03_{\pm 0.10}$ | $+\mathbf{1.03}_{\pm 0.09}$ |

seeds (more often for experiments with higher variability, see details in Appendix B), and we report the corresponding standard error. We bold the best-performing approach and any approach whose average performance plus standard error overlaps it.

**Hyper-parameters.** For all architectures and datasets, we set hyperparameters shared by all methods (e.g., learning rate) mostly to values cited in prior work [3] As explained in Secs. 2.2 and 2.3, the effectiveness of flat-minima optimizers is highly sensitive to their additional hyper-parameters. We select hyper-parameters using a grid search over a held-out validation set. Specifically, for SWA we follow Izmailov et al. [48] and hold the update frequency $\nu$ constant to once per epoch and tune the start time $E \in \{0.5T, 0.6T, 0.75T, 0.9T\}$ ($T$ is the number of baseline training epochs). Izmailov et al. [48] argue that a cyclical learning rate starting from $E$ helps to encourage exploration of the basin. For the sake of simplicity, we average the iterates of the baseline directly but include even earlier starting times (i.e., $0.5T, 0.6T$). For SAM, we tune its neighborhood size $\rho \in \{0.01, 0.02, 0.05, 0.1, 0.2\}$, as in previous work [22, 4].

Appendix B contains the values of all hyper-parameters and additional training details (including public model checkpoints, hardware infrastructure, software libraries, etc.) to ensure full reproducibility alongside open-sourcing our code.

## 4.1 Computer Vision

**Supervised Classification (SC).** We evaluate the CNN architectures WideResNets [99] with 28 layers and width 10, and PyramidNet (PN) with 110 layers and widening factor 272 [38] as well as Vision Transformer (ViT) [19] and MLP-Mixer [87] on CIFAR{10, 100} [63]. All experiments use basic data augmentations: horizontal flip, padding by four pixels, random crop, and cutout [17].

**Self-Supervised Learning (SSL).** We consider the following methods on CIFAR10 and ImageNette[4]: Momentum Contrast [41], a Simple framework for Contrastive Learning (SimCLR) [10], Simple Siamese representation learning (SimSiam) [11], Barlow Twins [100], Bootstrap your own Latent (BYOL) [35], and Swapping Assignments between multiple Views of the same image (SwAV) [7]. All SSL methods use a ResNet-18 [40] as backbone network. To test the frozen representations, we use $k$-nearest-neighbor classification with a memory bank [94]. We choose $k = 200$ and temperature $\tau = 0.1$ to reweight similarities. Compared to learning a linear model on top of the representations, this evaluation procedure is more robust to hyperparameter changes [59].

---

[3]Sometimes with minor modifications, e.g., adjusting per-device batch sizes to be compatible with our GPU infrastructure.

[4]`https://github.com/fastai/imagenette`

Figure 4: (a) NLP test results: Open-Domain Question Answering and Natural Language Understanding (GLUE) including paraphrase, sentiment analysis, and textual entailment. (b) GRL test results: Node Property Prediction (NPP), Graph Property Prediction (GPP), Link Property Prediction (LPP).

| Task | Model | Baseline | SWA | SAM | WASAM |
|---|---|---|---|---|---|
| NQ | FiD | $49.35_{\pm0.44}$ | $-0.20_{\pm0.33}$ | $+0.33_{\pm0.19}$ | $+0.48_{\pm0.21}$ |
| TriviaQA | FiD | $67.74_{\pm0.29}$ | $+0.40_{\pm0.24}$ | $+0.89_{\pm0.03}$ | $+0.92_{\pm0.10}$ |
| COLA | RoBERTa | $60.41_{\pm0.22}$ | $+0.09_{\pm0.08}$ | $+1.57_{\pm1.20}$ | $+1.41_{\pm1.14}$ |
| SST | RoBERTa | $94.95_{\pm0.13}$ | $-0.30_{\pm0.27}$ | $-0.23_{\pm0.40}$ | $+0.19_{\pm0.14}$ |
| MRPC | RoBERTa | $89.14_{\pm0.57}$ | $+0.08_{\pm0.49}$ | $+0.73_{\pm0.43}$ | $+0.81_{\pm0.38}$ |
| STSB | RoBERTa | $90.40_{\pm0.02}$ | $+0.00_{\pm0.05}$ | $+0.38_{\pm0.17}$ | $+0.35_{\pm0.16}$ |
| QQP | RoBERTa | $91.36_{\pm0.07}$ | $+0.01_{\pm0.06}$ | $+0.08_{\pm0.07}$ | $+0.06_{\pm0.08}$ |
| MNLI | RoBERTa | $87.41_{\pm0.09}$ | $+0.08_{\pm0.11}$ | $+0.39_{\pm0.02}$ | $+0.35_{\pm0.03}$ |
| QNLI | RoBERTa | $92.96_{\pm0.06}$ | $-0.08_{\pm0.11}$ | $+0.09_{\pm0.01}$ | $+0.11_{\pm0.06}$ |
| RTE | RoBERTa | $80.09_{\pm0.23}$ | $-0.23_{\pm0.20}$ | $+0.70_{\pm0.65}$ | $-0.46_{\pm0.12}$ |

(a)

| Task | Model | Baseline | SWA | SAM | WASAM |
|---|---|---|---|---|---|
| NPP: Proteins | SAGE | $77.79_{\pm0.18}$ | $-0.17_{\pm0.22}$ | $-0.02_{\pm0.13}$ | $-0.11_{\pm0.15}$ |
| | DGCN | $85.42_{\pm0.17}$ | $+0.11_{\pm0.08}$ | $-0.14_{\pm0.05}$ | $-0.08_{\pm0.07}$ |
| NPP: Products | SAGE | $78.92_{\pm0.08}$ | $+0.39_{\pm0.10}$ | $+0.13_{\pm0.08}$ | $+0.57_{\pm0.03}$ |
| | DGCN | $73.88_{\pm0.13}$ | $+0.44_{\pm0.14}$ | $+0.08_{\pm0.09}$ | $+0.53_{\pm0.05}$ |
| GPP: Code2 | GCN | $16.04_{\pm0.09}$ | $+0.73_{\pm0.11}$ | $+0.36_{\pm0.08}$ | $+0.93_{\pm0.15}$ |
| | GIN | $15.73_{\pm0.11}$ | $+0.83_{\pm0.11}$ | $+0.57_{\pm0.09}$ | $+1.10_{\pm0.09}$ |
| GPP: Molpcba | GIN | $28.10_{\pm0.11}$ | $+0.40_{\pm0.18}$ | $-0.33_{\pm0.14}$ | $+0.33_{\pm0.16}$ |
| | DGCN | $25.65_{\pm0.13}$ | $+1.90_{\pm0.20}$ | $-0.13_{\pm0.18}$ | $+1.34_{\pm0.12}$ |
| LPP: Biokg | CP | $84.06_{\pm0.00}$ | $+0.07_{\pm0.01}$ | $0.00_{\pm0.03}$ | $+0.08_{\pm0.02}$ |
| | ComplEx | $84.94_{\pm0.01}$ | $+0.14_{\pm0.01}$ | $-0.02_{\pm0.01}$ | $+0.12_{\pm0.02}$ |
| LPP: Citation2 | GCN | $79.52_{\pm0.41}$ | $-0.05_{\pm0.52}$ | $+1.32_{\pm0.06}$ | $+1.50_{\pm0.13}$ |
| | SAGE | $81.95_{\pm0.02}$ | $+1.15_{\pm0.02}$ | $-0.31_{\pm0.07}$ | $+0.86_{\pm0.04}$ |

(b)

## 4.2 Natural Language Processing

We consider the task of open domain question answering (ODQA) using a T5-based model Fusion-In-Decoder (FiD) [47]. We evaluate FiD-base on the test sets of Natural Questions (NQ) [64] and TriviaQA [50]. We also consider natural language understanding tasks included in the GLUE benchmark [91], which cover acceptability, sentiment, paraphrase, similarity, and inference. We fine-tune RoBERTa-base [72] for each task individually and report the results on the GLUE dev set.

## 4.3 Graph Representation Learning

We use a subset of the Open Graph Benchmark (OGB) datasets [45]. The tasks are node property prediction (NPP), graph property prediction (GPP), and link property prediction (LPP). For each task, we use two of the following GNN architectures and matrix factorization methods: GCN [57], DeeperGCN (DGCN) [68], SAGE [37], GIN [95], ComplEx [89], and CP [66]. We use popular training schemes, such as virtual nodes, cluster sampling [14], or relation prediction as auxiliary training objective [13]. The reported metrics are ROC-AUC for Proteins, Accuracy for Products, F1 score for Code2, Average precision for Molpcba, and Mean Reciprocal Rank for Biokg/Citation2.

## 4.4 9 Findings

We use $\mathcal{G}(\cdot)$ to describe the generalization accuracy/F1/ROCAUC/AP/MRR of all optimizers {Non-flat baseline(NF), SWA, SAM, WASAM}.

1. **Datasets matter.** For example, for node property prediction (Proteins), we see that no flat optimizer improves over the baseline optimizer; however, for (Products), flat-minima optimizers on the same architectures significantly improve over the baseline. We further explore the impact of different data augmentation strategies in Appendix A.7.

2. **Architectures matter,** e.g., there is a vast difference across model architectures for link property prediction on the Citation2 dataset: using a GNN with GCN layers, SAM achieves a statistically significant boost of >1.30% and SWA slightly hurts the performance. When we replace the GCN layers with SAGE layers (and fix everything else), we see a boost of $> 1.15\%$ for SWA, while SAM hurts the performance by $-0.31\%$.

3. **SWA underperforms on NLP tasks.** SAM achieves the best performance in 7/10 experiments on NLP tasks, consistent with the findings of Bahri et al. [4], which show that SAM can boost performance across a wide range of NLP tasks. However, SWA never performs best, only improving the results in 1/10 cases, and even hurting the performance on 4 tasks. Surprisingly, $\mathcal{G}(\text{WASAM}) > \mathcal{G}(\text{SAM})$ for {SST, QNLI} while SWA decreases performance in these cases.

4. **SWA beats SAM on GRL tasks.** $\mathcal{G}(\text{SWA}) > \mathcal{G}(\text{NF})$ in 10/12 experiments, while $\mathcal{G}(\text{SAM}) > \mathcal{G}(\text{NF})$ only in 4. We examine why SAM under-performs in Appendix A.4.

5. **SWA does not work well with Transformers.** SWA often does not improve and sometimes hurts performances, as can be seen in the ViT results in Table 2, and NLP results in Fig. 4a. In contrast, SAM has some positive effects in these settings. We explore this further in Appendix A.3.

6. **SWA and SAM improve SSL task performance.** This is non-trivial as the theoretical motivation behind finding flat minima is linked to supervised learning losses [43, 77, 21]. Concurrently to our work, Ramesh et al. [82] report that SAM helps for contrastive CLIP [79] models too.

7. **Flat optimizers do not strictly improve over non-flat optimizers.** The non-flat optimizer is nearly always the best for NPP Proteins and SSL methods on CIFAR10. We investigate the NPP Proteins solutions in the next subsection and recommend a more thorough investigation of the landscapes of SSL objectives for future work.

8. **Flat-minima optimizers offer asymmetric payoffs:** at worst, they decreased performance by $-0.30\%$, at best, they increased it by $2.60\%$.

9. **Averaging SAM iterates often improves over SWA or SAM alone**. $\mathcal{G}(\text{WASAM}) > \min(\mathcal{G}(\text{SWA}), \mathcal{G}(\text{SAM}))$ in 39/42 cases. We hypothesize that asymmetric payoffs are the reason: when either SWA or SAM does not improve over the baseline (as discussed above), it does not hurt (much) either, hence WASAM is more robust across all tasks.

### 4.5 Why do flat-minima optimizers fail?

Here, we audit one of the cases, where neither $\boldsymbol{\theta}^{\text{SWA}}$ nor $\boldsymbol{\theta}^{\text{SAM}}$ improves over $\boldsymbol{\theta}^{\text{NF}}$ (this happens in 3 out of 42 cases): training a GraphSAGE [37] model on OGB-Proteins: a protein-protein interaction graph where the goal is to predict the presence of protein functions (multi-label binary classification) [45]. $\boldsymbol{\theta}^{\text{SWA}}$ performs noticeably worse; $\boldsymbol{\theta}^{\text{SAM}}$ performs about equally well.

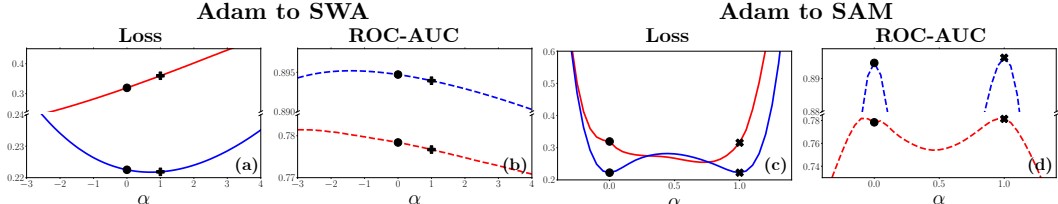

Figure 5: GraphSAGE on OGB-Proteins: Adam's ($\bullet$) solution performs about equally well as SAM ($\times$), and better than SWA ($+$).

Fig. 5 shows two linear interpolations: between $\boldsymbol{\theta}^{\text{NF}}$ (ADAM) and (1) $\boldsymbol{\theta}^{\text{SWA}}$ (Figures 5a and 5b), and (2) $\boldsymbol{\theta}^{\text{SAM}}$ (Figures 5c and 5d). In contrast to success cases in Fig. 2, here: (a) for both SWA and SAM, the training loss minimizer is very uncorrelated with the test loss minimizer; (b) SAM and ADAM seem to be contained in the same test loss/accuracy basin. More analyses can be found in the Appendix.

## 5 Limitations and Future Work

First, some of the fixed, shared hyperparameter values we used from previous works may harm the effect of flat optimizers. The ideal experimental design includes tuning all hyperparameters independently for the non-flat optimizer, SWA, SAM, and WASAM. However, this forces the number of required runs to grow exponentially in unique hyperparameters and quickly renders this benchmark infeasible.

Second, despite our best efforts to evaluate the optimizers on a broad range of benchmark tasks, there are still plenty of unexplored domains; especially some of which are known to be sensitive to careful optimization. For example, bi-level optimization problems [24] are common in generative modeling [31, 42], deep reinforcement learning [60, 44], meta-learning [81, 52], or causal machine learning [53, 54]. We are unaware of an investigation of flat minima optimization for such problems.

Third, in general, we believe fruitful directions of research include (a) optimizers that explicitly find basins where training loss flatness more directly corresponds to higher hold-out accuracy, (b) post-processing methods for existing optimization runs to move into flatter regions of these basins [2], (c) loss functions whose contours more tightly align with accuracy contours, (d) the study of flat-minima hyperparameter interactions (e.g., learning rate and neighborhood radius in SAM) (see Appendices A.5 and A.6 for first results), (e) analyses of flat minima optimization on convergence speed [51].

Our benchmark results point to which tasks would most benefit from improving these future work directions: graph learning tasks would benefit from improvements in (a), as SAM is never among the best-performing method, and language tasks would benefit if (b) is improved, as SWA is never among the best performing method).

## 6 Conclusion

We investigated when flat minima optimizers work by conducting a fair comparison of two popular flat-minima optimizers. We examined the behavior of SWA/SAM by analyzing their loss landscapes on two representative deep learning tasks. Our next step was to evaluate their generalization performance on a broad and diverse set of tasks (in data, learning settings, and model architectures). Based on this benchmarking, we identified 9 findings, of which some directly guide future work directions. Finally, when SWA/SAM did not improve over baselines, common assumptions seemed broken (i.e., train-to-test loss minimizers were not correlated).

## Acknowledgements

We are very grateful to Kilian Q. Weinberger and Gao Huang for initial discussions and intuition, Pontus Stenetorp for NLP experimental design advice, and Oscar Key for feedback on the draft. JK and LL acknowledge support by the Engineering and Physical Sciences Research Council with grant number EP/S021566/1. This research was supported through Azure resources provided by The Alan Turing Institute and credits awarded by Google Cloud.

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
