# A    Additional Analyses

## A.1    CKA Similarities

We compute the CKA [61, 96] and cosine similarities of network output logits on train and test set, respectively. Table 3 shows the results.

Table 3: **Pairwise CKA [61] and cosine similarities** between non-flat (NF) and SWA/SAM solutions. SWA solutions produce predictions more similar to NF ones than SAM.

| Task | $s_{\text{CKA}}(\boldsymbol{\theta}^{\text{NF}}, \boldsymbol{\theta}^{\text{SWA}})$ | $s_{\text{cosine}}(\boldsymbol{\theta}^{\text{NF}}, \boldsymbol{\theta}^{\text{SWA}})$ | $s_{\text{CKA}}(\boldsymbol{\theta}^{\text{NF}}, \boldsymbol{\theta}^{\text{SAM}})$ | $s_{\text{cosine}}(\boldsymbol{\theta}^{\text{NF}}, \boldsymbol{\theta}^{\text{SAM}})$ |
|---|---|---|---|---|
| WRN-CIFAR100 (Train) | 0.9880 | 0.9812 | 0.9810 | 0.9240 |
| WRN-CIFAR100 (Test) | 0.9137 | 0.9732 | 0.8580 | 0.9045 |
| GIN-Code2 (Train) | 0.8522 | 0.9730 | 0.7276 | 0.9515 |
| GIN-Code2 (Test) | 0.8677 | 0.9750 | 0.7275 | 0.9516 |
| RoBERTa-QNLI (Train) | 0.9997 | 0.9991 | 0.9790 | 0.9510 |
| RoBERTa-QNLI (Valid) | 0.9830 | 0.9959 | 0.9550 | 0.9530 |
| RoBERTa-RTE (Train) | 0.9931 | 0.9891 | 0.9831 | 0.9628 |
| RoBERTa-RTE (Test) | 0.9314 | 0.9567 | 0.8808 | 0.8927 |
| GIN-Molpcba (Train) | 0.8886 | 0.9973 | 0.7441 | 0.9804 |
| GIN-Molpcba (Test) | 0.8772 | 0.9942 | 0.7232 | 0.9730 |

The results show that the SAM solutions produce predictions that are less similar to the non-flat baseline than SWA solutions, as indicated by lower CKA and cosine similarities. This result is in line with Observation 1 and 2 from Sec. 3.1.

## A.2    Saddle point of SAM's GIN solution

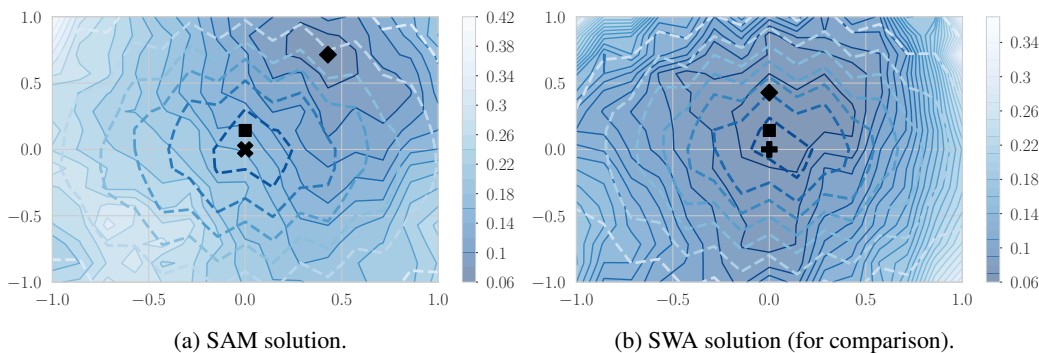

(a) SAM solution.                                            (b) SWA solution (for comparison).

Figure 6: **SAM and SWA solution in 2D random plane for GIN-Code2 task.** We depict $\boldsymbol{\theta}^{\text{SAM}}, \boldsymbol{\theta}^{\text{SWA}}$ by $\times, +$, respectively, the test set F1 score maximizer by $\blacksquare$, and the training loss minimizer by $\blacklozenge$. The converged SAM solution is distant from the training loss minimizer in the 2D plane: $\mathcal{L}_{\text{train}}(\boldsymbol{\theta}^{\text{SAM}}) = 0.1779 \gg 0.0672 = \mathcal{L}_{\text{train}}(\boldsymbol{\theta}^{\blacklozenge})$. Further, losses (—) and F1 scores (·····) are not well-aligned. In contrast, the SWA solution is almost the training loss minimizer: $\mathcal{L}_{\text{train}}(\boldsymbol{\theta}^{\text{SWA}}) = 0.0661 \approx 0.0609 = \mathcal{L}_{\text{train}}(\boldsymbol{\theta}^{\blacklozenge})$. Also, losses and F1 scores are better aligned. Yet, $\boldsymbol{\theta}^{\text{SAM}}$ and $\boldsymbol{\theta}^{\text{SWA}}$ perform about equally well on the test set, see Fig. 3d.

To gather further evidence on whether the SAM solution is a saddle point, we analyze its Hessian eigenvalue density, following Ghorbani et al. [29] and using the Stochastic Lanczos Quadrature algorithm [30]. Fig. 7 shows the density, including significant probability mass for both positive and negative eigenvalues, indicating that the solution is a saddle point.

## A.3    Why does SWA not work well on NLP tasks?

In Fig. 4a, we saw that SWA had only a mild effect on the generalization performance of NLP tasks, sometimes even decreasing it. Here, we seek to investigate why that is.

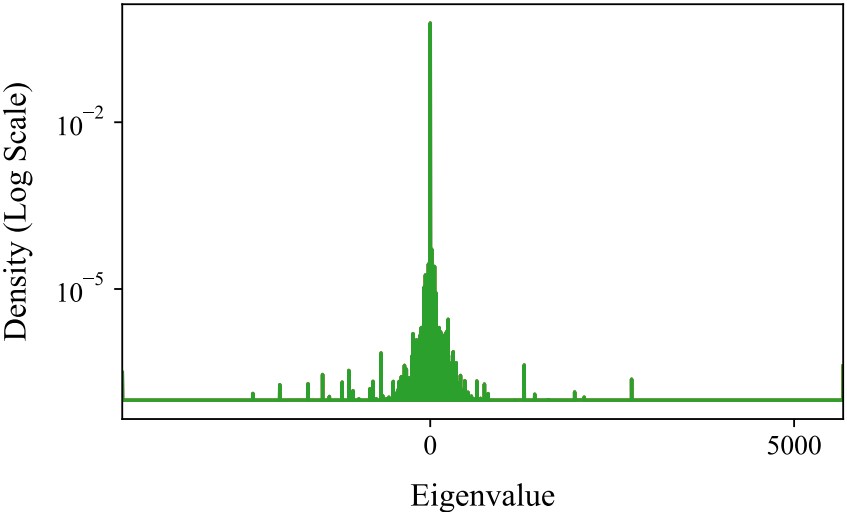

Figure 7: **Hessian Eigenvalue Density [29] of SAM's GIN-Code2 solution**: We observe significant probability mass around both positive and negative eigenvalues, indicating that this solution is a saddle point.

We consider two tasks: (i) the RTE task, for which SWA decreases the performance by around $-0.23_{\pm 0.20}$ compared to Adam, (ii) the QNLI task, for which SWA decreases the performance by $-0.08_{\pm 0.11}$. In both cases, SAM improved the performance statistically significantly over Adam.

For the QNLI task in Fig. 8a, we observe that SWA finds a lower/higher training loss/accuracy than Adam, respectively. However, the test loss/accuracy is higher/lower at the SWA solutions and the loss functions seem less well correlated in between both solutions (i.e, for $\alpha \in [0, 1]$).

For the RTE task in Fig. 8b, we note that SWA finds a solution that is closer to a sharply increasing side. This may happen if the baseline optimizer skips or goes around sharper solutions (e.g., due to large step sizes) and the average pulls it towards these suboptimal regions.

Further, in Table 3, we notice very high values of $s_{\text{CKA}}(\boldsymbol{\theta}^{\text{NF}}, \boldsymbol{\theta}^{\text{SWA}}), s_{\text{cosine}}(\boldsymbol{\theta}^{\text{NF}}, \boldsymbol{\theta}^{\text{SWA}})$ for both training and test sets, indicating that the predictions are indeed very similar. In contrast, $s_{\text{CKA}}(\boldsymbol{\theta}^{\text{NF}}, \boldsymbol{\theta}^{\text{SAM}}), s_{\text{cosine}}(\boldsymbol{\theta}^{\text{NF}}, \boldsymbol{\theta}^{\text{SAM}})$ is lower, especially for the test set.

### A.4 Why does SAM not work well on GRL tasks?

Fig. 10 shows the interpolations between the Adam and SAM solution. We do not observe a significant loss/accuracy difference between the two different basins. One possible explanation for this phenomenon is that the loss surface for this task is "globally well-connected" [96], yielding many basins with very similar geometric properties.

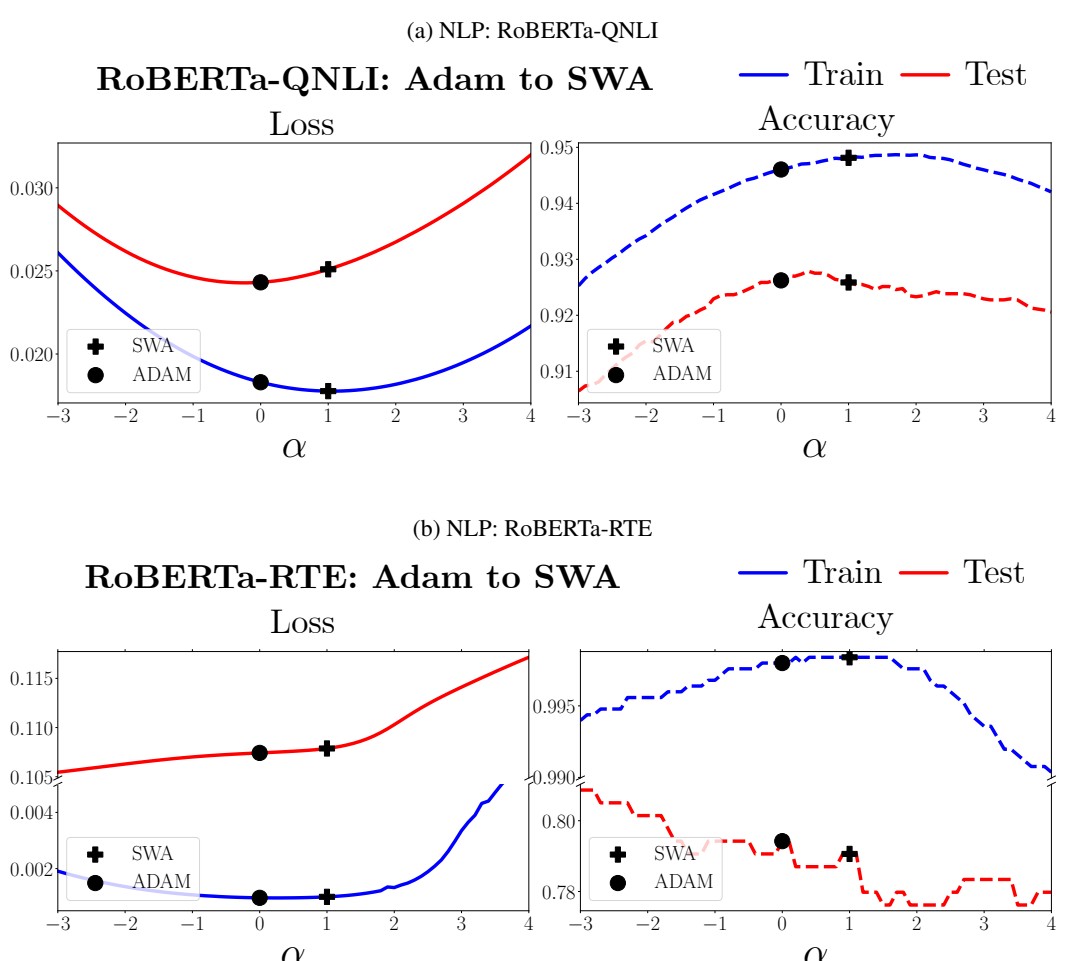

Figure 8: Training (blue) and test (red) losses (—) and accuracies (·····) of linear interpolations $\boldsymbol{\theta}(\alpha) = (1-\alpha)\boldsymbol{\theta} + \alpha\boldsymbol{\theta}'$ between Adam solutions ($\bullet, \alpha = 0.0$) and SWA ($+, \alpha = 1.0$).

Figure 9: GPP: Molpcba-GIN

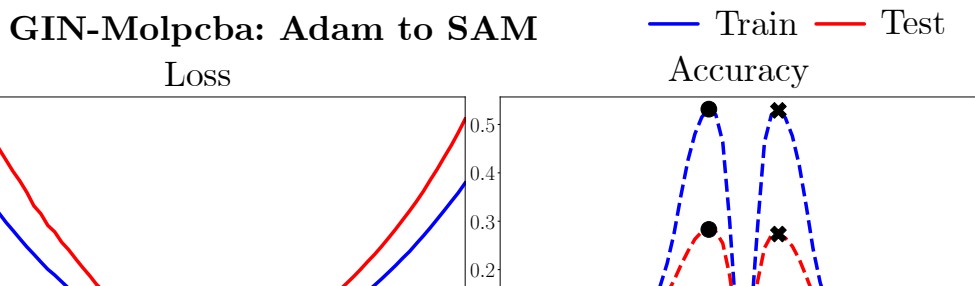

Figure 10: Training (blue) and test (red) losses (—) and accuracies (·····) of linear interpolations $\boldsymbol{\theta}(\alpha) = (1-\alpha)\boldsymbol{\theta} + \alpha\boldsymbol{\theta}'$ between Adam solutions ($\bullet, \alpha = 0.0$) and SAM ($\times, \alpha = 1.0$).

### A.5 Does changing SAM's $\rho$ result in different basins?

Here, we plot linear interpolations of solutions obtained by smaller and larger $\rho$ values. Overall, we find that all solutions seem to lie in different basins indicated by high loss barriers in between ($\alpha = 0.5$) them.

### A.5.1 WRN-28-10

For the WRN-28-10 model investigated in Sec. 3.1 and Sec. 4, we set $\rho = 0.1$ (as determined by hyper-parameter tuning on validation loss).

Figure 11: **WRN-28-10**: Changing SAM's $\rho$ result in different basins.

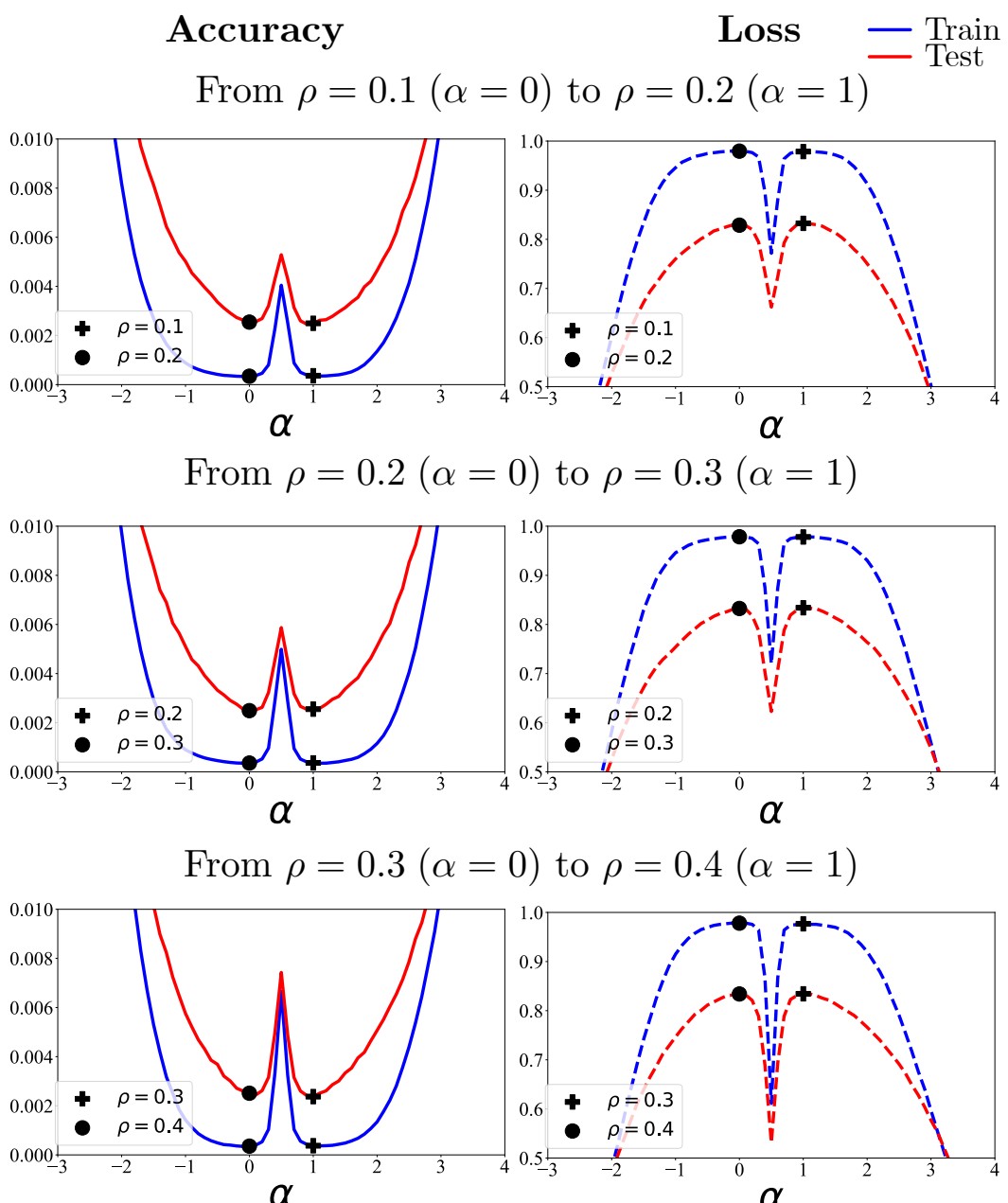

### A.5.2 GIN-Code2

For the GIN model investigated in Sec. 3.1 and Sec. 4, we set $\rho = 0.15$ (as determined by hyper-parameter tuning on validation loss).

Figure 12: **GIN-Code2**: Changing SAM's $\rho$ result in different basins.

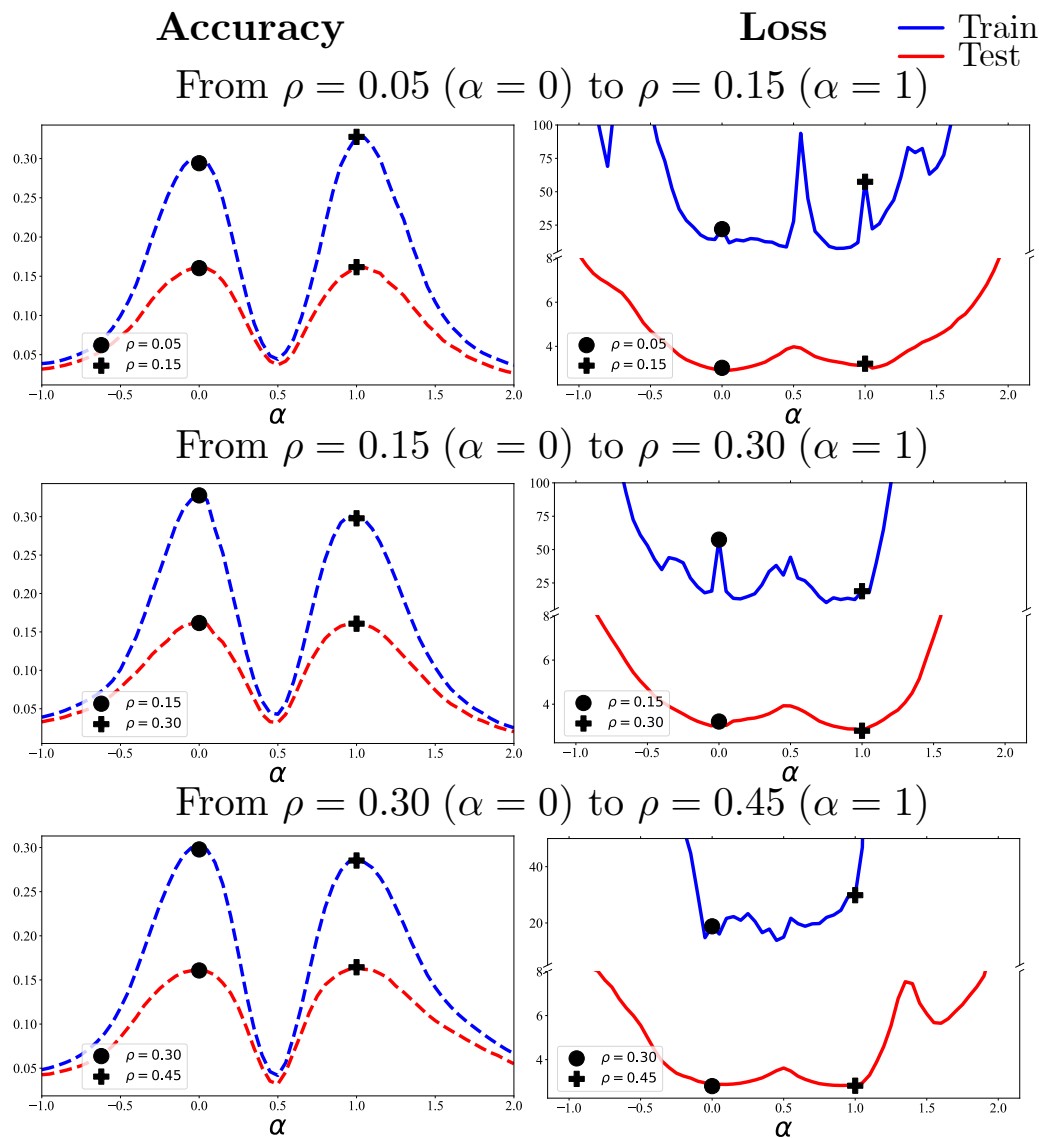

## A.6 Does changing the base optimizer impact SWA's and SAM's effectiveness?

The influence of the base optimizer on SWA/SAM's effectiveness is under-explored. In the previous experiments, we use the default base optimizers from existing code repositories or, as reported in previous works. Here, we want to conduct an initial investigation into its effect on SWA and SAM.

In the following experiments, we only switch the base optimizer and keep everything else fixed (including hyper-parameters such as learning rate, etc.). We train (i) a ResNet-34 (similar to WRN-28-10 but smaller) on CIFAR100, once per SGD with momentum and AdamW [73], and (ii) a GIN model on Code2, as in Sec. 3, but using RMSprop instead of Adam. We choose AdamW and RMSProp, since they are commonly used in image classification [73, 19, 88] and graph representation learning (GRL) [71, 70, 80], respectively.

Due to time constraints, we only report results obtained with one random seed (except for GIN-Code2 with Adam, which we already evaluated across three random seeds in our initial submission, see Fig. 4b). Further, again due to time constraints, for the ResNet-34 task, we do not conduct a hyper-parameter search of SAM's $\rho$ but set it to $\rho = 0.05$, as this value has been reported to be a good default value [22].

Table 4 show the test performances with switched base optimizers. First, discussing task (i), we note that AdamW under-fits the model and generalizes poorly compared to SGD. Here, SAM exacerbates the performance even further, performing even worse than the AdamW baseline. The reasons for that are unclear, and we leave an investigation into them for future work.

For task (ii) using RMSprop, we observe that both flat-minima optimizers improve over the baseline performance. However, compared to when using Adam, they perform even more similarly, with SAM only being $0.03\%$ better. Interestingly, the combination WASAM again performs best.

Table 4: **Test accuracies/F1 score of switched base optimizers**.

| Task | Baseline | SWA | SAM | WASAM |
|---|---|---|---|---|
| ResNet34 on CIFAR100 (SGD) | 76.14 | + 1.50 | +1.91 | **+ 2.60** |
| ResNet34 on CIFAR100 (AdamW) | 72.14 | **+ 0.57** | -2.29 | -1.11 |
| GIN on Code2 (Adam) | $15.73_{\pm 0.11}$ | $+ 0.83_{\pm 0.11}$ | $+ 0.57_{\pm 0.09}$ | $\mathbf{+ 1.10_{\pm 0.09}}$ |
| GIN on Code2 (RMSprop) | 15.30 | + 0.67 | + 0.70 | **+ 1.62** |

## A.7 Does changing the data augmentation impact SWA's and SAM's effectiveness?

In this ablation, we want to understand whether different amounts and data augmentation strategies impact SWA's or SAM's effectiveness. As an experimental setup, we consider training a ResNet18 on CIFAR100 for 200 epochs with SGD with a momentum of 0.9, initial learning rate 0.1, and cosine learning rate schedule. The three data augmentation strategies are (i) none, (ii) basic (random crop, random horizontal flipping), and (iii) AutoAugment following the CIFAR10 policy [15].

Table 5 shows the results. We find that with no data augmentation used, SWA, SAM and WASAM improve the baseline results by about the same amount, while SAM improves over SWA when data augmentation is used, and WASAM performs best.

Table 5: **Test accuracies of ResNet18 on Cifar100 with different data augmentation schemes**.

| Data Augmentation | Baseline | SWA | SAM | WASAM |
|---|---|---|---|---|
| None | 60.78 | + 2.23 | + 2.20 | **+ 2.24** |
| Basic | 76.40 | + 0.13 | + 0.74 | **+ 1.01** |
| AA [15] | 67.59 | + 3.03 | + 3.35 | **+ 4.17** |

## A.8 Does using a constant learning rate at the end of training improve SWA?

In this ablation, we aim to understand the impact of a constant learning rate at the end of the training, as originally suggested by [48]. We follow the same experimental setup as in Table 5 and choose the Basic Data Augmentation. At the last 25% of training (starting from epoch 150), we set the learning rate to 0.05. We find that this slightly worsens the SWA performance by $-0.66\%$ compared to running SWA without changing the learning rate schedule, as explained in Sec. 4.

# B   Experimental details

## B.1   Computer Vision

We mostly adopt the hyper-parameter values from Foret et al. [22] for WRN-28-10 and PyramidNet-272, from Dosovitskiy et al. [19] for ViT, and from [87] for MLP-Mixer models. We average all results across three random seeds.

### B.1.1   Supervised Classification

We train WideResNets [99] with 28 layers and width 10 (WRN28-10) and PyramidNet [38] with 110 layers and widening factor $\alpha = 272$ (PyramidNet-272) from scratch. The Vision Transformer (ViT) base model with input patch size 16 (ViT-B/16) and MLP-Mixer base model with input patch size 16 (MLP-Mixer-B/16) start from pre-trained checkpoints available at https://console.cloud.google.com/storage/vit_models/. The reason for using pre-trained checkpoints for the ViT and MLP-Mixer models is that, due to their lack of some inductive biases inherent to CNNs, such as translation equivariance and locality, they do not generalize well when trained on insufficient amounts of data [19]. Table 6 shows the hyper-parameters for each architecture.

Table 6: Hyper-parameters for Supervised Classification (SC): CIFAR-{10, 100} (Table 2)

| Hyper-Parameter | WRN28-10 | PyramidNet-272 | ViT-B/16 | MLP-Mixer-B/16 |
|---|---|---|---|---|
| Base Optimizer | SGD | SGD | SGD | SGD |
| Batch size | 256 | 256 | 100 | 170 |
| Data augmentation | | Inception-style + Cutout [17] | | |
| Dropout rate | | 0.0 | | |
| Epochs | 200 | 200 | – | – |
| Gradient clipping norm | – | – | 1.0 | 1.0 |
| Learning rate schedule | | cosine | | |
| Peak learning rate | 0.1 | 0.05 | 0.03 | 0.03 |
| Steps | – | – | 12500 | 12500 |
| SGD Momentum | | 0.9 | | |
| Warmup steps | – | – | 500 | 500 |
| Weight decay | $5e-4$ | $5e-4$ | 0.0 | 0.0 |
| **CIFAR-10** | | | | |
| SAM $\rho$ | 0.05 | 0.05 | 0.1 | 0.02 |
| Averaging start $E$ (SWA) | 60% | 60% | 75% | 90% |
| Averaging start $E$ (WASAM) | 90% | 75% | 75% | 90% |
| **CIFAR-100** | | | | |
| SAM $\rho$ | 0.1 | 0.1 | 0.2 | 0.05 |
| Averaging start $E$ (SWA) | 60% | 60% | 75% | 90% |
| Averaging start $E$ (WASAM) | 90% | 75% | 75% | 90% |

### B.1.2   Self-Supervised Learning

Table 7 shows the hyper-parameters for each SSL method. We use implementations from the lightly package, available at https://github.com/lightly-ai/lightly [86].

## B.2   Natural Language Processing

For the task of Open Domain Question Answering, we adapt the hyper-parameter values and the 25 retrieved passages for each question from 47. We report the Exact Match score of FiD-base model on Natural Questions (NQ) and TriviaQA test sets. For GLUE benchmark, we report Matthew's Corr for CoLA, Pearson correlation coefficient for STSB, and accuracy for the the rest of the datasets. Results are all evaluated on the dev set of GLUE benchmark. We use the RoBERTa-base as our backbone language model, implemented with Huggingface Transformers [92]. Most of the task-specific hyper-parameter values are adapted from 1.

## B.3   Graph Representation Learning

We mostly adapt the hyper-parameter values from Hu et al. [45] for GCN [57], SAGE [37], and GIN [95], from Chen et al. [13] for [66] and ComplEx [89], and from Li et al. [68] for DGCN. Due to high standard errors, we averaged the results of a few tasks more than three times, as mentioned in the following tables.

Table 7: Hyper-parameters for Self-Supervised Learning (SSL): CIFAR-10, ImageNette, results in (Table 2)

| Hyper-Parameter | MoCo | SimCLR | SimSiam | BarlowTwins | BYOL | SwaV |
|---|---|---|---|---|---|---|
| Backbone Network | | | | ResNet-18 | | |
| Base Optimizer | | | SGD | | | Adam |
| Data augmentation | | | SimCLR [10] | | | Multi-Crop [7] |
| Dropout rate | | | | 0.0 | | |
| Epochs | | | | 800 | | |
| Embedding dimensions | | | | 512 | | |
| KNN memory bank size | | | | 4096 | | |
| Learning rate schedule | | | | cosine | | |
| Peak learning rate | | | $6e-2$ | | | $1e-3$ |
| SGD Momentum | | | 0.9 | | | – |
| Weight decay | | | $5e-4$ | | | $1e-6$ |
| **CIFAR-10** | | | | | | |
| Batch size | | | | 512 | | |
| Crop size | | | – | | | 32 |
| Gaussian blur | | | | 0% | | |
| SAM $\rho$ | 0.01 | 0.01 | 0.01 | 0.05 | 0.01 | 0.05 |
| Averaging start $E$ (SWA) | 75% | 90% | 75% | 90% | 60% | 60% |
| Averaging start $E$ (WASAM) | 90% | 90% | 90% | 90% | 75% | 90% |
| **ImageNette** | | | | | | |
| Batch size | | | | 256 | | |
| Crop size | | | – | | | 128, 64 |
| Gaussian blur | | | | 50% | | |
| SAM $\rho$ | 0.01 | 0.01 | 0.02 | 0.05 | 0.05 | 0.01 |
| Averaging start $E$ (SWA) | 50% | 90% | 75% | 75% | 90% | 50% |
| Averaging start $E$ (WASAM) | 50% | 50% | 90% | 75% | 90% | 50% |

Table 8: Hyper-parameters for NPP tasks, results in Fig. 4b.

| Hyper-Parameter | SAGE | DGCN |
|---|---|---|
| **NPP: OGB-Proteins** | | |
| Aggregation method | Mean | Softmax |
| Base optimizer | Adam | Adam |
| Convolution layer | SAGE | DyResGEN |
| Dropout rate | 0.0 | 0.1 |
| Hidden dimensions | 256 | 64 |
| Learning rate | 0.01 | 0.001 |
| Normalization layer | – | Layer norm |
| Number of epochs | 2000 | 1000 |
| Number of layers | 3 | 112 |
| Number of random seeds | 5 | 3 |
| Training cluster number | 1 | 15 |
| Weight decay | 0.0 | 0.0 |
| SAM $\rho$ | 0.01 | 0.02 |
| Averaging start $E$ (SWA) | 90% | 90% |
| Averaging start $E$ (WASAM) | 90% | 90% |
| **NPP: OGB-Products** | | |
| Aggregation method | Mean | Softmax |
| Base optimizer | Adam | Adam |
| Batch size | 20000 | – |
| Convolution layer | SAGE | Gen |
| Dropout rate | 0.5 | 0.5 |
| Evaluation cluster number | – | 8 |
| Learning rate | 0.01 | 0.001 |
| Hidden dimensions | 256 | 128 |
| Normalization layer | – | Batch norm |
| Number of epochs | 30 | 50 |
| Number of layers | 3 | 14 |
| Number of random seeds | 5 | 3 |
| Training cluster number | – | 10 |
| Weight decay | 0.0 | 0.0 |
| SAM $\rho$ | 0.01 | 0.02 |
| Averaging start $E$ (SWA) | 90% | 60% |
| Averaging start $E$ (WASAM) | 75% | 90% |

Table 9: Hyper-parameters for GPP: OGB-Code2, results in Fig. 4b.

| Hyper-Parameter | GCN | GIN |
|---|---|---|
| **GPP: OGB-Code2** | | |
| Aggregation method | Mean | Mean |
| Base optimizer | Adam | Adam |
| Batch size | 128 | 128 |
| Convolution layer | GCN | GIN |
| Dropout rate | 0.0 | 0.0 |
| Learning rate | 0.001 | 0.001 |
| Hidden dimensions | 300 | 300 |
| Normalization layer | Batch norm | Batch norm |
| Number of random seeds | 3 | 3 |
| Number of epochs | 15 | 30 |
| Number of layers | 5 | 5 |
| Virtual node embeddings | True | True |
| Vocabulary size | 5000 | 5000 |
| Weight decay | 0.0 | 0.0 |
| SAM $\rho$ | 0.2 | 0.15 |
| Averaging start $E$ (SWA) | 50% | 50% |
| Averaging start $E$ (WASAM) | 50% | 50% |

Table 10: Hyper-parameters for GPP: OGB-Molpcba, results in Fig. 4b.

| Hyper-Parameter | GIN | DGCN |
|---|---|---|
| **GPP: OGB-Molpcba** | | |
| Aggregation method | Mean | Mean |
| Batch size | 512 | 512 |
| Base optimizer | Adam | Adam |
| Convolution layer | GIN | GEN |
| Dropout rate | 0.0 | 0.2 |
| Learning rate | 0.001 | 0.001 |
| Normalization layer | Batch norm | Batch norm |
| Number of epochs | 100 | 50 |
| Number of layers | 5 | 14 |
| Number of random seeds | 3 | 3 |
| Hidden dimensions | 300 | 256 |
| Virtual node embeddings | False | True |
| Weight decay | 0.0 | 0.0 |
| SAM $\rho$ | 0.01 | 0.15 |
| Averaging start $E$ (SWA) | 90% | 75% |
| Averaging start $E$ (WASAM) | 90% | 50% |

Table 11: Hyper-parameters for LPP: OGB-Biokg, results in Fig. 4b.

| Hyper-Parameter | CP | ComplEx |
|---|---|---|
| **GPP: OGB-Biokg** | | |
| Base optimizer | Adam | Adam |
| Batch size | 500 | 500 |
| Learning rate | 0.1 | 0.1 |
| Number of random seeds | 3 | 3 |
| Number of epochs | 30 | 50 |
| Rank | 1000 | 1000 |
| Regularizer | N3 | N3 |
| Weight decay | 0.0 | 0.0 |
| SAM $\rho$ | 0.1 | 0.05 |
| Averaging start $E$ (SWA) | 50% | 50% |
| Averaging start $E$ (WASAM) | 90% | 50% |

Table 12: Hyper-parameters for LPP: OGB-Citation2, results in Fig. 4b.

| Hyper-Parameter | GCN | SAGE |
|---|---|---|
| **GPP: OGB-Citation2** | | |
| Aggregation method | Mean | Mean |
| Base optimizer | Adam | Adam |
| Batch size | 256 | 512 |
| Convolution layer | GCN | SAGE |
| Dropout rate | 0.0 | 0.2 |
| Hidden dimensions | 256 | 256 |
| Number of epochs | 300 | 300 |
| Number of layers | 3 | 3 |
| Number of random seeds | 3 | 3 |
| Normalization layer | – | – |
| Learning rate | 0.001 | 0.0005 |
| Virtual node embeddings | False | False |
| Weight decay | 0.0 | 0.0 |
| SAM $\rho$ | 0.02 | 0.01 |
| Averaging start $E$ (SWA) | 75% | 90% |
| Averaging start $E$ (WASAM) | 60% | 90% |