# OpenReview forum: "When Do Flat Minima Optimizers Work?"
_NeurIPS.cc/2022/Conference — NeurIPS 2022 Accept_

### Official Review · Reviewer_ssYQ · 2022-07-09

**Rating:** 7
**Confidence:** 4
**Soundness:** 4 excellent
**Presentation:** 4 excellent
**Contribution:** 4 excellent

**Summary:**

This paper analyses the properties of two flat minima optimizers, Stochastic Weight Averaging (SWA) and Sharpness-Aware Minimization (SAM). The nature of the minima reached by the two methods is studied through linear interpolations and dominant Hessian eigenvalues. SWA and SAM are benchmarked across a wide range of tasks in Natural language understanding, Graph Representation Learning, and Image Classification. Based on these results, a set of observations about SWA and SAM are described.

**Questions:**

1. Including observations relating to the robustness of the optimizer to its hyperparameters would also add practical value to the work. The role of the hyperparameters of the flat-optimizer in reaching certain minima can also be studied. For example, Does a large $\rho$ lead to a completely different basin than a small $\rho$?
2. It would be clearer if the authors mentioned the tasks/models where a pretrained model is used. For example, it is mentioned in the Appendix that the CIFAR-10 ViT results used a pretrained ImageNet ViT. The characteristics of the flat optimizers might be very different if a model is trained from scratch with the flat optimizer instead of being finetuned with the flat optimizer.
3. It is mentioned that one of the reasons why flat optimizers fail is because the train and test minimizers are not correlated. Can this hypothesis be more thoroughly evaluated by plotting the linear interpolation plots on an OOD test set?. For example, for the CIFAR-10 dataset, one could evaluate on CIFAR-10.1/ CINIC-10 dataset.
4. Question 3 (d) of the checklist has been answered: "Yes, See Appendix." But the details regarding the compute are not available in the Appendix.
5. Why does Algorithm 1 returns $\theta^{SWA+SAM}$? Should it not return  $\theta^{SWA}$

**Limitations:**

The limitations of the work have been described in detail.

**Strengths And Weaknesses:**

Strengths:
1. The analysis of the minima found by SAM and SWA has not been studied earlier. The observations are quite interesting and valuable to the community.
2. The authors perform a detailed comparison of SAM and SWA across a wide range of benchmarks. All the results are reported with error bars.
3. Overall, the paper is well written, with novel observations clearly stated and explained with supportive evidence.

Weaknesses:
1. This work has not studied the role of the base optimizer. The authors have compared SAM and SWA across multiple tasks and architectures. However, some methods use SGD, while others use Adam as the base optimizer. The loss landscape and generalization have been compared with SGD and Adam [1]. The comparison of SGD vs. Adam for the same task with SWA and SAM, respectively, would help disentangle the impact of the base optimizer.

2. Regarding Observation 3 (L166): "We visualize 2D plots moving along random directions (not shown here due to space)". This plot must be added to the Appendix along with the observations. Otherwise, the saddle point hypothesis is not backed up with evidence. Additionally, it would be good to plot the Eigen Spectral Density [2] and show the presence of significant negative eigenvalues, which can confirm the presence of saddle point.

[1] Zhou, P., Feng, J., Ma, C., Xiong, C., Hoi, S. C. H., et al. Towards theoretically understanding why sgd generalizes better than adam in deep learning. Advances in Neural Information Processing Systems, 33:21285–21296, 2020.

[2] Ghorbani, Behrooz, Shankar Krishnan, and Ying Xiao. "An investigation into neural net optimization via hessian eigenvalue density." International Conference on Machine Learning. PMLR, 2019.

---

> ### Author Response · Authors · 2022-08-02
> **Thank you very much for this insightful review! Response 1/2**
>
> Thanks a lot for your encouraging comments and careful feedback. We respond to your comments below and describe how we will update the work to reflect them.
>
> ### [”Weakness 1. This work has not studied the role of the base optimizer. The authors have compared SAM and SWA across multiple tasks and architectures. However, some methods use SGD, while others use Adam as the base optimizer. The loss landscape and generalization have been compared with SGD and Adam [1]. The comparison of SGD vs. Adam for the same task with SWA and SAM, respectively, would help disentangle the impact of the base optimizer.”]
>
> ### [1] Zhou, P., Feng, J., Ma, C., Xiong, C., Hoi, S. C. H., et al. Towards theoretically understanding why sgd generalizes better than adam in deep learning. Advances in Neural Information Processing Systems, 33:21285–21296, 2020.
>
> Here’s our selection strategy: for each task, we selected the base optimizer that was used in the most recent literature for that task. For instance, for image classification tasks and most SSL tasks, we use SGD with momentum, while for SwaV (SSL), NLP, and GRL tasks, Adam seems to be the go-to optimizer. Once we’ve selected the base optimizer, SAM will also use the same optimization strategy (but will, of course, compute gradients differently), as is done in prior work [2,3,4]. That said, we completely agree that it is interesting to understand how changing the base optimizer may change SWA’s and SAM’s effectiveness, holding the task fixed. We will run this and include it in the final version for a subset of the tasks. Thanks for pointing this out!
>
> [2] Ramesh, A., Dhariwal, P., Nichol, A., Chu, C., & Chen, M. (2022). Hierarchical text-conditional image generation with clip latents. *arXiv preprint arXiv:2204.06125*
>
> [3] Bahri, D., Mobahi, H., & Tay, Y. (2021). Sharpness-aware minimization improves language model generalization. *arXiv preprint arXiv:2110.08529*
>
> [4] Chen et al, When Vision Transformers Outperform ResNets without Pre-training or Strong Data Augmentations, ICLR 2022.
>
> ### [”Weakness 2. Regarding Observation 3 (L166): "We visualize 2D plots moving along random directions (not shown here due to space)". This plot must be added to the Appendix along with the observations. Otherwise, the saddle point hypothesis is not backed up with evidence. Additionally, it would be good to plot the Eigen Spectral Density [2] and show the presence of significant negative eigenvalues, which can confirm the presence of saddle point.”]
>
> ### [2] Ghorbani, Behrooz, Shankar Krishnan, and Ying Xiao. "An investigation into neural net optimization via hessian eigenvalue density." International Conference on Machine Learning. PMLR, 2019.
>
> 2D plot: We agree and added it to the revised version in the supplementary material, see A.2 of the revised paper in the supplementary material.
>
> Eigen Spectral Density [2] plot: We appreciate this suggestion! We will include Eigen spectral density plots in the final version. Thank you for pointing us to this technique.
>
> ## Answers to **Questions:**
>
> ### [”1. Including observations relating to the robustness of the optimizer to its hyperparameters would also add practical value to the work. The role of the hyperparameters of the flat-optimizer in reaching certain minima can also be studied. For example, Does a large ρ lead to a completely different basin than a small ρ?”]
>
> Thank you for this question. In our evaluation setup, we selected hyperparameters that minimized validation loss in order to reproduce the models that would be used by practitioners. As you suggest, we suspect that results will vary noticeably as hyperparameters change, e.g., if SAM’s radius $\rho$ is too small, it will likely find sharper minima, or if SWA’s starting epoch is very late, its solution will be almost identical to the non-flat one (also likely sharper). That said, if there’s an experiment you’d like to see investigated, we are happy to include this.
>
> ### [”2. It would be clearer if the authors mentioned the tasks/models where a pretrained model is used. For example, it is mentioned in the Appendix that the CIFAR-10 ViT results used a pretrained ImageNet ViT. The characteristics of the flat optimizers might be very different if a model is trained from scratch with the flat optimizer instead of being finetuned with the flat optimizer.”]
>
> You are right; this is important to note in the main text and describe any differences w.r.t. non-pre-trained models. We will add this to the final version.
>
> ### [”3. It is mentioned that one of the reasons why flat optimizers fail is because the train and test minimizers are not correlated. Can this hypothesis be more thoroughly evaluated by plotting the linear interpolation plots on an OOD test set?. For example, for the CIFAR-10 dataset, one could evaluate on CIFAR-10.1/ CINIC-10 dataset.”]
>
> Interesting idea! We are happy to include these linear interpolation plots for novel tasks such as OOD generalization.

---

> > ### Author Response · Authors · 2022-08-02
> > **Thank you very much for this insightful review! Response 2/2**
> >
> > ### [”4. Question 3 (d) of the checklist has been answered: "Yes, See Appendix." But the details regarding the compute are not available in the Appendix.”]
> >
> > Thank you for catching this! We ran all experiments on a single-GPU device, either an NVIDIA RTX 3090 or NVIDIA V100. We will add this information to the Appendix. Thank you again!
> >
> > ### [”5. Why does Algorithm 1 returns $\theta^{\text{SWA+SAM}}$? Should it not return $\theta^{\text{SWA}}$?”]
> >
> > Sorry, this is a typo! You are right, it should return $\theta^{\text{SWA}}$. We will correct this.

---

> > > ### Comment · Reviewer_ssYQ · 2022-08-06
> > > **Response to Author**
> > >
> > > I thank the authors for the detailed response to the review.
> > > 1. The additional CKA analysis between Non-Flat Optimizers and SAM/SWA is quite interesting and gives further evidence of the observations reported in the paper.
> > > 2. I thank the authors for adding the saddle point plot. However, I am not completely sure if the figure implies a saddle point for the converged SAM solution. Like I mentioned, the presence of negative eigenvalues at a low loss value could provide solid evidence of a saddle point. Also, is the "saddle point" observed in SAM only with GIN-Code2 Task? Or do the other tasks have similar phenomena?
> > > 3. Regarding Q1, a simple experiment that can be done is to analyze the solutions (with linear interpolations), of SAM from $\rho$ = 0.05 to $\rho$ = 0.5. Is it possible that the solution for $\rho$ = 0.1 and $\rho$ = 0.2, may lie in the same basin?
> > > 4. The paper would be strengthened if the authors add the comparison of base optimizer and the eigenvalue analysis in the final version.

---

> > > > ### Author Response · Authors · 2022-08-09
> > > > **We addressed your remaining questions in the revised Appendix (supplementary material).**
> > > >
> > > > Thank you so much for engaging with us in this discussion. We appreciate it very much.
> > > >
> > > > We just updated the Appendix (see supplementary material) and tried to address all your remaining questions.
> > > >
> > > > Addressing your above points:
> > > >
> > > > 2. In A.2, we added the Hessian eigenvalue density, see Figure 7. Significant probability mass on both negative and positive eigenvalues indicate that this solution is a saddle point.
> > > > 3. In A.5, we additionally trained six models and six linear interpolations.
> > > > 	* **WRN-28-10**: $\rho=\{0.1, 0.2, 0.3, 0.4\}$ with linear interpolations between
> > > >         * $\rho=0.1$ and $\rho=0.2$
> > > >         * $\rho=0.2$ and $\rho=0.3$
> > > >         * $\rho=0.3$ and $\rho=0.4$
> > > > 	* **GIN-Code2**: $\rho=\{0.05, 0.15, 0.30, 0.45\}$ with linear interpolations between
> > > >         * $\rho=0.05$ and $\rho=0.15$
> > > >         * $\rho=0.15$ and $\rho=0.30$
> > > >         * $\rho=0.30$ and $\rho=0.45$
> > > >     * The linear interpolations indicate that the considered solutions lie in different basins.
> > > >
> > > > 4. In A.6, we additionally trained six models and switched the base optimizer as follows:
> > > > * **ResNet-34 on CIFAR100** with both SGD (default choice for this task) and AdamW. Due to time constraints, we had to use the smaller ResNet-34 architecture; it is similar to WRN-28-10 but smaller. Using SGD as baseline (i.e., without SWA or SAM) improves over AdamW as the baseline by 4.00%. Interestingly, when using AdamW, SAM exacerbates the performance even further, performing even worse than the AdamW baseline. However, SWA improves over AdamW by 0.57%.
> > > > * **GIN on Code2** with RMSprop instead of Adam (default choice). Both flat-minima optimizers improve over the baseline performance. However, compared to Adam, they perform even more similarly, with SAM only being 0.03% better. Interestingly, the combination SWA+SAM again performs best.
> > > >
> > > > We hope this clarifies some of your concerns about our submission.
> > > > Thank you very much for helping us to improve the paper.

---

### Official Review · Reviewer_ah85 · 2022-07-10

**Rating:** 7
**Confidence:** 5
**Soundness:** 3 good
**Presentation:** 4 excellent
**Contribution:** 3 good

**Summary:**

The authors perform a detailed comparison of stochastic weight averaging (SWA) and sharpness-aware minimization (SAM), first with loss landscape analysis and then with a broad set of benchmarks. The authors find several surprising results, and in particular show that combining the two methods, it is often possible to achieve improvements over both of them, leading to consistently good performance.

**Questions:**

**Q1**. Could you please comment on the base optimizer used in each data domain (Vision, NLP, GRL)? Could the SWA poor performance on NLP tasks compared to Vision be explained by the different choice of base optimizer?

**Q2**. Could you pleas comment on W1? How is  $\theta^{NF}$ computed for both SWA and SAM? What is $\|\theta^{NF} - \theta^{SWA}\|$ and $\|\theta^{NF} - \theta^{SAM}\|$ in Fig. 2?

**Q3**. Could you pleas comment on W3? Why did you choose not to use a cyclical or constant learning rate schedule for SWA? Would it be possible to evaluate SWA with a constant learning rate schedule after epoch $E$ for a subset of your benchmarks to understand whether the choice of the learning rate schedule affected your results?

**Limitations:**

The authors adequately discuss the limitations of their work.

**Strengths And Weaknesses:**

**S** = strength, **W** = weakness, **Q** = question.

**S1**. The authors perform a detailed benchmark study that was missing from the literature. It is interesting to see that SAM and SWA are in fact quite complimentary, without one method clearly dominating the other.

**S2**. The benchmarks considered by the authors are quite diverse, including multiple domains. Moreover, the results differ between the different domains, showing the importance of a diverse benchmark.

**S3**. The combination SWA+SAM seems to be quite useful for practitioners potentially, as it works consistently well across a  broad set of benchmarks, while SWA and SAM are somewhat less consistent.

**S3**. The authors perform loss surface analysis experiments to provide intuition behind the results.

**W1**. In section 3.1 it is not clear how exactly the solutions $\theta^{NF}$ are obtained for each optimizer. My guess is that for SWA the authors used the same optimization trajectory but without averaging to obtain  $\theta^{NF}$, while for SAM they used an independent run? In this case, the relation between $(\theta^{NF}, \theta^{SWA})$ and the relation between $(\theta^{NF}, \theta^{SAM})$ will be quite different. In particular, I would expect the distance spanned by the horizontal access in Figure 2 to be much larger for SAM compared to SWA, making the interpretation of the plots more nuanced.

**W2**. In lines 226-227 the authors mention that for SWA they simply average the iterates of the baseline optimization run, without modifying the learning rate schedule. While this is a valid approach, the authors of SWA emphasize the importance of using a non-standard learning rate schedule (see e.g. [the official blogpost](https://pytorch.org/blog/pytorch-1.6-now-includes-stochastic-weight-averaging/#is-this-just-averaged-sgd)). The deviation from the standard SWA recipy makes the benchmarking results a bit harder to interpret.

---

> ### Author Response · Authors · 2022-08-02
> **Thank you very much for this insightful review!**
>
> Thank you very much for your insightful ideas on experimental design and for your positive feedback. We respond to your comments below and detail how we will update the paper to reflect these.
>
> ### [“W1.  In section 3.1 it is not clear how exactly the solutions $\theta^{\text{NF}}$ are obtained for each optimizer. My guess is that for SWA the authors used the same optimization trajectory but without averaging to obtain θNF, while for SAM they used an independent run? In this case, the relation between ($\theta^{\text{NF}}$,$\theta^{\text{SWA}}$)  and the relation between ($\theta^{\text{NF}}$,$\theta^{\text{SAM}}$) will be quite different. In particular, I would expect the distance spanned by the horizontal access in Figure 2 to be much larger for SAM compared to SWA, making the interpretation of the plots more nuanced.”]
>
> ### [“**Q2**. Could you please comment on W1? How is θNF computed for both SWA and SAM? What is $| \theta^{\text{NF}} - \theta^{\text{SWA}}|$  and $| \theta^{\text{NF}} - \theta^{\text{SAM}}|$ in Fig. 2?”]
>
> Yes you’re right: $\theta^{\text{NF}}$ is the solution obtained by the base optimizer, while $\theta^{\text{SWA}}$ is the averaged solution along the same trajectory. $\theta^{\text{SAM}}$ is indeed obtained through a different run. However, we made sure to start it from the same random initialization as $\theta^{\text{NF}}$. You’re absolutely right that the distance from $\theta^{\text{NF}}$ to $\theta^{\text{SAM}}$ is usually much larger than from $\theta^{\text{NF}}$ to $\theta^{\text{SWA}}$.
>
> Thank you for bringing this up! We agree it would be useful to include these solution distances. We have included CKA and cosine similarities in the revised Appendix A.1. We will add more distances, like the $L_1$ and $L_2$ distance, in the final version.
>
> ### [”**W2**. In lines 226-227 the authors mention that for SWA they simply average the iterates of the baseline optimization run, without modifying the learning rate schedule. While this is a valid approach, the authors of SWA emphasize the importance of using a non-standard learning rate schedule (see e.g. [the official blogpost](https://pytorch.org/blog/pytorch-1.6-now-includes-stochastic-weight-averaging/#is-this-just-averaged-sgd)). The deviation from the standard SWA recipy makes the benchmarking results a bit harder to interpret.”]
> ### [”**Q3**. Could you please comment on W2? Why did you choose not to use a cyclical or constant learning rate schedule for SWA? Would it be possible to evaluate SWA with a constant learning rate schedule after epoch E for a subset of your benchmarks to understand whether the choice of the learning rate schedule affected your results?”]
>
> Thank you for this question. We initially experimented with a cyclical and constant learning rate schedule for SWA and found that this worsened the performance compared to continuing with the default learning rate schedules. [1] similarly confirm empirically that the constant learning rate works better than the cyclical. That said, we are happy to include evaluations of SWA with cyclical learning rate for a subset of the experiments to show the effect of the learning rate schedule!
>
> [1] Cha et al, Swad: Domain generalization by seeking flat minima, NeurIPS 2021.
>
> ### [”**Q1**. Could you please comment on the base optimizer used in each data domain (Vision, NLP, GRL)? Could the SWA poor performance on NLP tasks compared to Vision be explained by the different choice of base optimizer?”]
>
> Here’s our selection strategy: for each task, we selected the base optimizer that was used in the most recent literature for that task. For instance, for image classification tasks and most SSL tasks, we use SGD with momentum, while for SwaV (SSL), NLP, and GRL tasks, Adam seems to be the go-to optimizer. Once we’ve selected the base optimizer, SAM will also use the same optimization strategy (but will, of course, compute gradients differently), as is done in prior work [2,3,4].
>
> We don’t believe that Adam especially harms the performance of SWA for NLP tasks, as the same overall update is also applied to SAM. That said, we completely agree that it is interesting to understand how changing the base optimizer may change SWA’s and SAM’s effectiveness, holding the task fixed. We will run this and include it in the final version for a subset of the tasks. Thanks for pointing this out!
>
> [2] Ramesh, A., Dhariwal, P., Nichol, A., Chu, C., & Chen, M. (2022). Hierarchical text-conditional image generation with clip latents. *arXiv preprint arXiv:2204.06125*
>
> [3] Bahri, D., Mobahi, H., & Tay, Y. (2021). Sharpness-aware minimization improves language model generalization. *arXiv preprint arXiv:2110.08529*
>
> [4] Chen et al, When Vision Transformers Outperform ResNets without Pre-training or Strong Data Augmentations, ICLR 2022.

---

> > ### Comment · Reviewer_ah85 · 2022-08-07
> > **Thank you for the rebuttal**
> >
> > Dear authors, thank you for the rebuttal and clarifications!
> >
> > > That said, we are happy to include evaluations of SWA with cyclical learning rate for a subset of the experiments to show the effect of the learning rate schedule!
> >
> > Constant learning rate could be sufficient, given the results in the work you mention and the SWA blogpost. I think it would be especially interesting to see these results for (1) vision problems (as the SWA was evaluated more extensively there and the learning rate schedule modification was found useful) and (2) nlp problems where you found SWA to perform poorly.
> >
> > I am happy with the response, and recommend an accept!

---

### Official Review · Reviewer_Twme · 2022-07-11

**Rating:** 5
**Confidence:** 5
**Soundness:** 3 good
**Presentation:** 3 good
**Contribution:** 3 good

**Summary:**

This paper focuses on two famous flat-minima optimizers, SAM and SWA. They provide lots of experiment results to analyze the solution of these two optimizers and find that they focus on different aspects. Finally, they try to combine them to get better performance than SAM or SWA and conduct some experiments in a variety of tasks to help us better understand these two algorithms.

**Questions:**

1. In the experiments, whether the distributed setting is used. The main reason is that we find this setting is very important for SAM, which may also affect the analysis of SAM.

2. We know that data augmentation can improve the generalization of neural networks. When using data augmentation, the improvement of SAM will be reduced. What is the relationship between data augmentation, SAM, and SWA.



**Limitations:**

The paper provides some experiment analysis to illustrate the properties of SAM and SWA.

1. The contribution of this paper. Combining SAM and SWA is not novel for me.

2. The performance of SWA+SAM is not very great for me. In some tasks, the performance are very close between SAM+SWA and SAM/SWA and in some experiments SAM+SWA will hurt the performance.

3. The experiment results of ViT training on ImageNet. I think this experiment is very important for SAM since SAM can achieve about 5% improvement on ImageNet.

**Strengths And Weaknesses:**

Strengths:

1. This paper provides lots of experiment results in various tasks to analyze SAM and SWA, which help us better understand them.

2. The proposed method is very clear and simple to reproduce.

3. SWA+SAM can achieve better performance in some tasks.

4. The figures and writing are very easy to understand.

Weakness:

1. Although the paper provides lots of experiment results, some important experiments are not included in this paper. For example, some related work illustrate that SAM can achieve a huge improvement in ViT training on ImageNet (5% improvement). I think this is very important for people to better understand the relationship between SAM and SWA.

2. Some comparison with current state of the art methods, such as GSAM.

3. The accuracy of SAM+SWA is not better than vanilla SAM in some experiments. In addition, in some tasks the performance are very close. I think that will also limit the contribution of this paper.

4. About the novelty of this paper. Although the paper focus on a very important problem, but the proposed method try to combine current two methods, SAM and SWA, which will also weak the contribution of this paper.

---

> ### Author Response · Authors · 2022-08-02
> **Thank you very much for this constructive review! Response 1/2**
>
> Thank you so much for your helpful and constructive review! We really appreciate your time and efforts. We respond to your comments below, describing how we will update the paper based on them.
>
> ## [“Weakness 1. …SAM can achieve a huge improvement in ViT training on ImageNet (5% improvement). I think this is very important for people to better understand the relationship between SAM and SWA.”]
> [”Limitation 3. The experiment results of ViT training on ImageNet. I think this experiment is very important for SAM since SAM can achieve about 5% improvement on ImageNet.”]
>
> Thank you for this comment. We assume you describe training from scratch on ImageNet, as most ViT papers use this strategy. We agree that this could be interesting.
>
> We have tried to calculate a ballpark cost estimate for this training below. Based on this, one training run of ViT-L/16 model on ImageNet from scratch is estimated to cost ~\\$5520. Since SAM takes twice the training time (due to two forward/backward passes per parameter update step), it should cost roughly twice this for the SAM training run. Further, in our evaluation protocol, we run hyper-parameter tuning as well as at least three random seeds for each experiment. We will try our best to get the resources allocated required to follow our experimental protocol, but this may take some time.
>
> ### Training Cost Estimation:
>
> For our estimation, we use [1], which directly applied SAM to [2]’s training pipeline, and used 128 TPU v3 cores. However, they do not mention the duration of the training and do not compare it against SWA.
>
> To estimate this, we use the figure reported in the original ViT paper [2]: 230 TPUv3-core-days (number of TPU v3 cores multiplied by the training time in days). As far as we know, this is the smallest of all mentioned training times. Because performance on TPU types scales linearly [3], 230 TPUv3-core-days are equivalent to 7.1875 days of training, with one instance being equipped with 32 v3 cores. One hour of such an instance costs \\$32 / hour [4], so we conclude that 7.1875 days * 24 hours * \\$32/hour = \\$5520 per training run. Note that this excludes the time needed to set up the instance.
>
> [1] Chen et al, When Vision Transformers Outperform ResNets without Pre-training or Strong Data Augmentations, ICLR 2022.
>
> [2] Dosovitskiy et al, An Image is Worth 16x16 Words: Transformers for Image Recognition at Scale, ICLR 2021.
>
> [3] [https://cloud.google.com/tpu/docs/regions-zones#experimenting](https://cloud.google.com/tpu/docs/regions-zones#experimenting)
>
> [4] [https://cloud.google.com/tpu/pricing](https://cloud.google.com/tpu/pricing)
>
> ## [”Weakness 2. Some comparison with current state of the art methods, such as GSAM.”]
>
> We opted against including a comparison with more recent SAM / SWA variants for the following reason: there’s no clear winner in either the SAM and SWA categories (we are aware of 11 very recent SAM variants (ASAM, ESAM, GSAM, SSAM, SAF, MESA, Fisher SAM, LookSAM, Look-LayerSAM, $\delta$-SAM, Generalized SAM [5]) and 4 SWA variants (FGE, SWAD, TWA, EMA)). This is because there are inconsistencies in experimental design, and a broader benchmark beyond image classification tasks is missing. This means we would need to compare against a large subset of them, some of which require additional hyper-parameters to tune (e.g., GSAM’s neighborhood radius schedule $\rho_{min}, \rho_{max}$ or FGE’s $\alpha_1, \alpha_2$). The analysis we do in this paper would be infeasible if we included more methods: e.g., including the same pairwise comparisons of methods requires a quadratic increase in experiments (i.e., $m(m - 1)/2$, where $m = $# of methods).
>
> Further, in our initial experiments, we tried comparing with ASAM. However, we found that the results were nearly identical to SAM, sometimes slightly better. Because of this, we decided to focus on a more detailed analysis of the most well-known methods, SAM and SWA.
>
> [5] Zhao et al, Penalizing Gradient Norm for Efficiently Improving Generalization in Deep Learning, ICML 2022.

---

> > ### Author Response · Authors · 2022-08-02
> > **Thank you very much for this constructive review! Response 2/2**
> >
> > ### [”Weakness 3. The accuracy of SAM+SWA is not better than vanilla SAM in some experiments. In addition, in some tasks the performance are very close. I think that will also limit the contribution of this paper.”]
> > ### [”Weakness 4. About the novelty of this paper. Although the paper focus on a very important problem, but the proposed method try to combine current two methods, SAM and SWA, which will also weak the contribution of this paper.”]
> > ### [”Limitation 1. The contribution of this paper. Combining SAM and SWA is not novel for me.”]
> > ### [“Limitation 2. The performance of SWA+SAM is not very great for me. In some tasks, the performance are very close between SAM+SWA and SAM/SWA and in some experiments SAM+SWA will hurt the performance.”]
> >
> > We believe there is a slight confusion here; the novelty of our approach is not purely the mixture of SWA and SAM but our analyses of their similarities/differences and their conclusions. Only one of our conclusions was that mixing could help generalization in some cases. We aim to take a step back and analyze existing methods more thoroughly before proposing yet another algorithm.
> >
> > As mentioned by Reviewers ah85 and ssYQ, the novelty of this work is in the observations that resulted from comparing SWA and SAM (in Sections 3.1 and 4.4). Most flat-minima papers published in the last two years are extensions of SAM. Yet, none of them compares against SWA, an older, computationally cheaper, and arguably simpler method. This is a troubling trend because, e.g., SAM has little to even a negative impact on the performance of GRL tasks, while SWA often improves over the baseline. Further, most of these works evaluate their performance on a small subset of image classification tasks, leaving the question open whether their claimed superiority over original SAM is actually robust across a broad range of tasks.
> >
> > Our paper establishes a more thorough testbed for future optimizers (we will open-source the code to reproduce all experiments, see supplementary material). We hope this creates a standard for future work evaluating flat-minima optimizers.
> >
> > ## Answers to **Questions:**
> >
> > ### [”1. In the experiments, whether the distributed setting is used. The main reason is that we find this setting is very important for SAM, which may also affect the analysis of SAM.”]
> >
> > Good question. Here we do not consider the distributed version of SAM. For each optimization step, we compute both the perturbation and parameter update step on a single device. We agree this is important to investigate in future work once the non-distributed setting is well understood.
> >
> > ### [”2. We know that data augmentation can improve the generalization of neural networks. When using data augmentation, the improvement of SAM will be reduced. What is the relationship between data augmentation, SAM, and SWA.”]
> >
> > Thank you for this question! We have not explored this direction yet, but we are happy to add this interesting ablation experiment for some of the image classification tasks to the final version!

---

> > > ### Comment · Reviewer_Twme · 2022-08-07
> > > **Thanks for your response**
> > >
> > > Thanks for your response.
> > >
> > > I understand the training cost of ViT training. I think someone can help you explore it in the future.
> > >
> > > I still think distributed setting is very important and maybe you can add the analysis about that in the future.
> > >
> > > I will keep my score.

---

> > > > ### Comment · Reviewer_ah85 · 2022-08-07
> > > > **Re: SAM for VIT on ImageNet1k**
> > > >
> > > > I wanted to mention the recent paper [1], which as far as I know, achieves competitive results to SAM with standard Adam optimization, they argue that the original ImageNet1k ViT baselines were undertuned significantly.
> > > >
> > > > I agree that including some results for the distributed setting for SAM would be quite interesting. Note that it is possible to run these experiments on one GPU, you do not need to actually run distributed experiments. I believe it can improve results in some cases (see $m$-*sharpness* in the original SAM paper).
> > > >
> > > > [1] *Better plain ViT baselines for ImageNet-1k*;
> > > > Lucas Beyer, Xiaohua Zhai, Alexander Kolesnikov
> > > > https://arxiv.org/abs/2205.01580

---

> > > > > ### Comment · Reviewer_Twme · 2022-08-08
> > > > > **Thanks for your comments**
> > > > >
> > > > > Thanks for your comments.
> > > > >
> > > > > I know [1] can further improve the performance of SAM (about 79% for ViT), but I still have several concerns:
> > > > >
> > > > > - (1) [1] tries to tune the structure of the original ViT model to obtain a better performance, and I also think SAM can achieve better performance on this new ViT model. So I don't think you can say " achieves competitive results to SAM with standard Adam optimization";
> > > > >
> > > > > - (2) [1] also uses strong data augmentation (such as AutoAug or RandAug) to obtain a similar performance as the original SAM. When we also use the strong augmentation for SAM, it can also further improve the accuracy (from 79+\% to 81\%). That is also a great achievement. So I also don't think standard Adam can achieve this performance.
> > > > >
> > > > > In my experience, for small datasets (such as CIFAR), we can also obtain a great performance without distributed setting.
> > > > >
> > > > > However, for large-scale datasets (such as ImageNet) and large models (such as ViT), SAM cannot work well without distributed setting. So I think distributed setting is very important and it could be better if the authors could provide some results about that.

---

> > > > > ### Author Response · Authors · 2022-08-08
> > > > > **Thank you for your engagement in the discussion!**
> > > > >
> > > > > We thank you both, reviewer ah85 and Twme, for your response and for engaging with us in the discussion. We very much appreciate it.
> > > > >
> > > > > Regarding the distributed setting experiments: As you mentioned, the m-sharpness analysis in the original SAM paper points out that for smaller values of m, generalization can improve. We want to point out the concurrent work [1], which explicitly studies m-sharpness in more detail, empirically and theoretically, and aims to explain its success.
> > > > >
> > > > > Concerning our paper, are there any particular experiments or analyses you would like to see that have not been discussed by [1] yet?
> > > > >
> > > > > [1] Towards Understanding Sharpness-Aware Minimization, Andriushchenko and Flammarion, ICML 2022, https://arxiv.org/pdf/2206.06232.pdf

---

### Official Review · Reviewer_J6KH · 2022-07-11

**Rating:** 5
**Confidence:** 4
**Soundness:** 2 fair
**Presentation:** 3 good
**Contribution:** 2 fair

**Summary:**

This paper compares two well-known flat-minima-seeking methods, stochastic weight averaging (SWA) and sharpness aware minimization (SAM). Based on the geometric loss landscape analysis, the paper suggests combining SWA and SAM. Through experiments conducted on a wide range of tasks, the paper attempts to address when and why each method fails.

**Questions:**

- Do SWA solutions and non-flat baseline solutions share optimization trajectories? If so, observation 1 seems to be a naïve consequence.
- In Section 3, is Adam optimizer used for SWA and SAM in GIN/Code2 task? Or, only for non-flat solutions?
- Regarding line 296, I cannot find an explanation in the Appendix as to why SWA and transformers do not work well together.


**Limitations:**

The paper provided limitations along with future research directions in Section 5.


**Strengths And Weaknesses:**

**Strengths**
- The paper tackles a significant and practically-relevant problem supported by a fair amount of experiments conducted across various tasks and domains. There is a need for comparative studies of flat-minima-seeking algorithms.
- Overall, the paper is well-written and easy to follow.

**Weaknesses**

*(W1) Robustness to various settings:*
- In Section 3, it is unclear whether the findings in Figure 2 can be generalized to other settings. For instance, is the result robust to varying hyperparameters (e.g. SAM's radius) or even to different random seeds? Considering that this is an analysis paper, the robustness of the findings should be highlighted.
- Although the authors stated the scope of the work in Section 2.4, it is questionable whether the results would hold under variants of SWA and SAM (e.g., Adaptive SAM, Fast Geometric Ensemble).

*(W2) Credibility of linear interpolation analysis:*
- Regarding observations 3 and 4 in Section 3.1, one may consider asymmetric valleys [1]. While SAM solutions are closer to the sharper basin in certain directions, it is not guaranteed to be true in all directions.
- I cannot find a 2D plot in the Appendix, which was mentioned in line 168.
- It remains unclear how SAM+SWA solutions escape the saddle point in Fig 3.

*(W3) Novelty of proposed algorithm:*
- The proposed way of mixing SWA and SAM outperforms baselines in half of the proposed cases. In this regard, it lacks novelty.
- Further, the motivation, i.e., observation 4, needs to be enhanced in line with the previous comment (W2). For example, I wonder if dominant Hessian eigenvalue analysis would hold under different settings since it is well-known that such analysis is not translation invariant w.r.t. neural network parameters.

*(W4) Lack of interpretation:*
- While the findings in Section 4.4 are interesting, it is mostly devoted to when SWA and SAM fail, not why. In order to conduct future research based on the paper’s observations, well-reasoned interpretations or hypotheses are crucial.
- As in A.2, a study concerning global flatness and well-connectedness or other distance metrics (e.g., CKA) would be worthwhile [2].

**References**\
[1] He, Haowei, Gao Huang, and Yang Yuan. "Asymmetric valleys: Beyond sharp and flat local minima." Advances in neural information processing systems 32 (2019). \
[2] Yang, Yaoqing, et al. "Taxonomizing local versus global structure in neural network loss landscapes." Advances in Neural Information Processing Systems 34 (2021): 18722-18733.

------------------------------------------------
(Post-rebuttal) I have updated my score to 5.

---

> ### Author Response · Authors · 2022-08-02
> **Thank you very much for this constructive review! Response 1/3**
>
> Thank you very much for your detailed feedback and time reviewing the work. We respond to your comments and indicate how we will update our paper accordingly below. We include references in the last comment 3/3.
>
> ## *(W1) Robustness to various settings:*
>
> ### [“In Section 3, it is unclear whether the findings in Figure 2 can be generalized to other settings. For instance, is the result robust to varying hyperparameters (e.g. SAM's radius) or even to different random seeds? Considering that this is an analysis paper, the robustness of the findings should be highlighted.”]
>
> Thank you for this question. In our evaluation setup, we selected hyperparameters that minimized validation loss in order to reproduce the models that would be used by practitioners. We suspect that results will vary noticeably as hyperparameters change, e.g., if SAM’s radius is too small, it will likely find sharper minima, or if SWA’s starting epoch is very late, its solution will be almost identical to the non-flat one (also likely sharper). That said, if there’s an experiment you’d like to see investigated, we are happy to include this.
>
> We also agree that it is interesting to understand how results change across different random seeds. Initially, we tested this on some models and noticed that they look (almost) identical. However, we will run this again more carefully and include it in the final version.
>
> ### [“Although the authors stated the scope of the work in Section 2.4, it is questionable whether the results would hold under variants of SWA and SAM (e.g., Adaptive SAM, Fast Geometric Ensemble).”]
>
> We opted against including a comparison with more recent SAM / SWA variants for the following reason: there’s no clear winner in either the SAM and SWA categories (we are aware of 11 very recent SAM variants (ASAM, ESAM, GSAM, SSAM, SAF, MESA, Fisher SAM, LookSAM, Look-LayerSAM, $\delta$-SAM, Generalized SAM [9]) and 4 SWA variants (FGE, SWAD, TWA, EMA)). This is because there are inconsistencies in experimental design, and a broader benchmark beyond image classification tasks is missing. This means we would need to compare against a large subset of them, some of which require additional hyper-parameters to tune (e.g., GSAM’s neighborhood radius schedule $\rho_{min}, \rho_{max}$ or FGE’s $\alpha_1, \alpha_2$). The analysis we do in this paper would be infeasible if we included more methods: e.g., including the same pairwise comparisons of methods requires a quadratic increase in experiments (i.e., $m(m - 1)/2$, where $m = $# of methods).
>
> Further, in our initial experiments, we tried comparing with ASAM. However, we found that the results were nearly identical to SAM, sometimes slightly better. Because of this, we decided to focus on a more detailed analysis of the most well-known methods, SAM and SWA.
>
> ## *(W2) Credibility of linear interpolation analysis:*
>
> ### [“Regarding observations 3 and 4 in Section 3.1, one may consider asymmetric valleys [1]. While SAM solutions are closer to the sharper basin in certain directions, it is not guaranteed to be true in all directions.”]
>
> It’s true that SAM solutions may not be sharper in other directions. However, we agree with [1], which pointed out that sharpness in a single direction can harm generalization and can sometimes be improved through averaging.
>
> ### [”I cannot find a 2D plot in the Appendix, which was mentioned in line 168.”]
>
> Thanks for pointing this out! We originally had the 2D plot but forgot to include it in the final version. Thanks for catching this! We added it to the revised version in the supplementary material, see A.2.
>
> ### [”It remains unclear how SAM+SWA solutions escape the saddle point in Fig 3.”]
>
> This is because SAM+SWA averages iterates before the saddle point (if they were all in the saddle point, the solution would remain there). We agree it is important to state this explicitly; we’ll do this in the final version.

---

> > ### Author Response · Authors · 2022-08-02
> > **Thank you very much for this constructive review! Response 2/3**
> >
> > ## *(W3) Novelty of proposed algorithm:*
> >
> > ### [”The proposed way of mixing SWA and SAM outperforms baselines in half of the proposed cases. In this regard, it lacks novelty.”]
> >
> > We believe there is a slight confusion here; the novelty of our approach is not purely the mixture of SWA and SAM but our analyses of their similarities/differences and their conclusions. Only one of our conclusions was that mixing could help generalization in some cases. We aim to take a step back and analyze existing methods more thoroughly before proposing yet another algorithm.
> >
> > As mentioned by Reviewers ah85 and ssYQ, the novelty of this work is in the observations that resulted from comparing SWA and SAM (in Sections 3.1 and 4.4). Most flat-minima papers published in the last two years are extensions of SAM. Yet, none of them compares against SWA, an older, computationally cheaper, and arguably simpler method. This is a troubling trend because, e.g., SAM has little to even a negative impact on the performance of GRL tasks, while SWA often improves over the baseline. Further, most of these works evaluate their performance on a small subset of image classification tasks, leaving the question open whether their claimed superiority over original SAM is actually robust across a broad range of tasks.
> >
> > Our paper establishes a more thorough testbed for future optimizers (we will open-source the code to reproduce all experiments, see supplementary material). We hope that this creates a standard for future work evaluating flat-minima optimizers.
> >
> > ### [”Further, the motivation, i.e., observation 4, needs to be enhanced in line with the previous comment (W2). For example, I wonder if dominant Hessian eigenvalue analysis would hold under different settings since it is well-known that such analysis is not translation invariant w.r.t. neural network parameters.”]
> >
> > We agree that using Hessian-based metrics to measure flatness is not perfect. The reason we included it is because we train all methods using the same initialization, controlling for spurious factors such as translation. Further, it is widely used [2,3,4,5], making it more familiar and potentially easier to interpret in the optimization community. If there are other sharpness metrics you’d like to see, we would be happy to include them.
> >
> >
> >
> > ## *(W4) Lack of interpretation:*
> >
> > ### [”While the findings in Section 4.4 are interesting, it is mostly devoted to when SWA and SAM fail, not why. In order to conduct future research based on the paper’s observations, well-reasoned interpretations or hypotheses are crucial.”]
> >
> > We agree. This is why in Section 4.5, we started to investigate why observation 6 holds in some settings. We found that one crucial assumption is broken: the shift between train and test is not a simple translation, as described in previous papers (e.g., asymmetric valleys [1]). Instead, they have different shapes, which means that flatter regions in training loss no longer correspond to lower test loss. We are interested in further investigating explanations for our other observations and hope to do this in future work.
> >
> > ### [“As in A.2, a study concerning global flatness and well-connectedness or other distance metrics (e.g., CKA) would be worthwhile [2].”]
> >
> > Thank you for this suggestion. We have added CKA measures of the two tasks in Section 3 and the failure case tasks to the revised supplementary material, see A.1.
> >
> > We are slightly confused about what you mean by “global flatness”; in [2], they only describe methods for local flatness (i.e., flatness within a certain neighborhood in parameter space).
> >
> > We actually are inspired by [2]’s measure of local flatness: [2] computes dominant Hessian eigenvalues (like we do in Section 3.3.) and Hessian trace. They show that both quantities are highly correlated and lead to the same conclusions (see e.g. Figures 2 and 3 in [2]).
> >
> > For well-connectedness, we opted against using Bezier curves as in [2], and instead used linear interpolations as in [6,7,8]. This is because compared with Bezier curves, linear interpolations do not need (i) additional hyper-parameters and (ii) a training procedure while still allowing one to estimate well-connectedness.
> >
> > Thank you for bringing this up. Please let us know if you have more requests for us to compute certain measures, we are happy to include them!

---

> > > ### Author Response · Authors · 2022-08-02
> > > **Thank you very much for this constructive review! Response 3/3**
> > >
> > > ## Answers to Questions
> > >
> > > ### [”Do SWA solutions and non-flat baseline solutions share optimization trajectories? If so, observation 1 seems to be a naïve consequence.”]
> > >
> > > Yes, the SWA solution uses the same trajectory as the non-flat baseline. We believe this observation is interesting due to the following, admittedly subtle, reason. During the course of non-flat optimization, iterates are likely to enter different basins due to the significant nonconvexity of NN training. If we average iterates in different basins, the average will not necessarily fall in the same basin as the final non-flat solution. Our observation is that when we tune SWA’s starting epoch to minimize validation loss, we always (in all the plots we visualized) average iterates in the same basin as the non-flat solution so that the SWA solution lies in the same basin as the non-flat solution. This means that, for some tasks where SWA outperforms the non-flat solution, there are better-generalizing solutions than the non-flat ones in the same basin. This motivates future work exploring other post-processing/tuning methods for converged non-flat solutions to improve generalization. We agree that this is not clear in the submitted manuscript, and we will improve this in the next version; thank you for that suggestion.
> > >
> > > ### [”In Section 3, is Adam optimizer used for SWA and SAM in GIN/Code2 task? Or, only for non-flat solutions?”]
> > >
> > > We always use the non-flat baseline as the base optimizer for SWA/SAM, across all experiments.
> > >
> > > ### [”Regarding line 296, I cannot find an explanation in the Appendix as to why SWA and transformers do not work well together.”]
> > >
> > > This is in Section A.1 “Why does SWA not work well on NLP tasks?” of the submitted paper (A.3 in the revised version). Here we show additional loss landscape visualizations of SWA using the RoBERTA transformer architecture in Figure 6. Different from Figure 3, in Figure 6, we note that the training accuracy is a poor surrogate of test accuracy (particularly for 6b). This means that finding flatter regions of training loss (a reasonable surrogate of training accuracy) does not lead to lower test accuracy. We will add this description to the appendix.
> > >
> > > # References
> > >
> > > [1] He, Haowei, Gao Huang, and Yang Yuan, Asymmetric valleys: Beyond sharp and flat local minima, NeurIPS 2019.
> > >
> > > [2] Yang, Yaoqing, et al., Taxonomizing local versus global structure in neural network loss landscapes, NeurIPS 2021.
> > >
> > > [3] Foret et al, Sharpness-aware Minimization for Efficiently Improving Generalization, ICLR 2021.
> > >
> > > [4] Chen et al, When Vision Transformers Outperform ResNets without Pre-training or Strong Data Augmentations, ICLR 2022.
> > >
> > > [5] Zhuang et al, Surrogate Gap Minimization Improves Sharpness-Aware Training, ICLR 2022.
> > >
> > > [6] Frankle et al, Linear mode connectivity and the lottery ticket hypothesis, ICML 2020.
> > >
> > > [7] Mirzadeh et al, Linear Mode Connectivity in Multitask and Continual Learning, ICLR 2021.
> > >
> > > [8] Juneja et al, Linear Connectivity Reveals Generalization Strategies, 2022, arXiv:2205.12411.
> > >
> > > [9] Zhao et al, Penalizing Gradient Norm for Efficiently Improving Generalization in Deep Learning, ICML 2022.

---

> > > > ### Comment · Reviewer_J6KH · 2022-08-06
> > > > **Thank you for your response.**
> > > >
> > > > Thank you for the detailed response!
> > > >
> > > > (W1, W2) Great, I believe incorporating the arguments above will strengthen the final version. Especially, the experiment settings should be clearly stated.
> > > >
> > > > (W3) Including eigenvalue spectral density plots may help.
> > > >
> > > > (W4 & questions) Great, I enjoyed reading the additional analysis.
> > > >
> > > > The score has been updated. I believe the paper should be accepted if all the comments are properly addressed in the final manuscript.

---

### Author Response · Authors · 2022-08-02
**Thank you for the constructive reviews!**

Thank you to all the reviewers for your very insightful comments and constructive reviews! We deeply appreciate your valuable time and efforts spent on improving our paper. In spite of the paper's imperfections, we are happy to see that all reviewers recognized the significance and value of the investigation of SWA and SAM for the community. Please see our reply to each reviewer individually below.

---

### Meta-Review · Area_Chair_f1cm · 2022-08-20

**Recommendation:** Accept
**Confidence:** Certain

**Metareview:**

This paper is an empirical investigation comparing two popular optimization techniques, namely SAM and SWA. Authors compare the performance of these two methods over a wide range of tasks and architectures. They also inspect and compare the properties of minima found by these methods. Given the existing interest in the ML community to understand these methods, reviewers are in agreement that insights from these paper are valuable and impactful enough to accept this paper.

**Award:**

No

---

### Decision · Program_Chairs · 2022-09-14

Accept